# The Power of Predictions in Online Control

**Chenkai Yu**
IIIS, Tsinghua University
yck17@mails.tsinghua.edu.cn

**Guanya Shi**
CMS, Caltech
gshi@caltech.edu

**Soon-Jo Chung**
CMS, GALCIT, JPL, Caltech
sjchung@caltech.edu

**Yisong Yue**
CMS, Caltech
yyue@caltech.edu

**Adam Wierman**
CMS, Caltech
adamw@caltech.edu

## Abstract

We study the impact of predictions in online Linear Quadratic Regulator control with both stochastic and adversarial disturbances in the dynamics. In both settings, we characterize the optimal policy and derive tight bounds on the minimum cost and dynamic regret. Perhaps surprisingly, our analysis shows that the conventional greedy MPC approach is a near-optimal policy in both stochastic and adversarial settings. Specifically, for length-$T$ problems, MPC requires only $O(\log T)$ predictions to reach $O(1)$ dynamic regret, which matches (up to lower-order terms) our lower bound on the required prediction horizon for constant regret.

## 1 Introduction

This paper studies the effect of using predictions for online control in a linear dynamical system governed by $x_{t+1} = Ax_t + Bu_t + w_t$, where $x_t$, $u_t$, and $w_t$ are the state, control, and disturbance (or exogenous input) respectively. At each time step $t$, the controller incurs a quadratic cost $c(x_t, u_t)$. Recently, considerable effort has been made to leverage and integrate ideas from learning, optimization and control theory to study the design of optimal controllers under various performance criteria, such as static regret [2, 3, 12, 13, 15, 20, 29], dynamic regret [16, 23] and competitive ratio [17, 28]. However, the study of online convergence when incorporating predictions has been largely absent.

Indeed, a key aspect of online control is considering the amount of available information when making decisions. Most recent studies focus on the basic setting where only past information, $x_0, w_0, \cdots, w_{t-1}$, is available for $u_t$ at every time step [2, 13, 15, 28]. However, this basic setting does not effectively characterize situations where we have accurate predictions, e.g., when $x_0, w_0, \cdots, w_{t-1+k}$ are available at step $t$. These types of accurate predictions are often available in many applications, including robotics [8, 27], energy systems [30], and data center management [22]. Moreover, there are many practical algorithms that leverage predictions, such as the popular Model Predictive Control (MPC) [6–9, 18, 19].

While there has been increased interest in studying online guarantees for control with predictions, to our knowledge, there has been no such study for the case of a finite-time horizon with disturbances. Several previous works studied the economic MPC problem by analyzing the asymptotic performance without disturbances [6, 7, 18, 19]. Rosolia and Borrelli [25, 26] studied learning for MPC but focused on the episodic setting with asymptotic convergence guarantees. Li et al. [23] considered a linear system where finite predictions of costs are available, and analyzed the dynamic regret of their new algorithm; however, they neither consider disturbances nor study the more practically relevant MPC approach. Goel and Hassibi [16] characterized the offline optimal policy (i.e., with infinite predictions) and cost in LQR control with i.i.d. zero-mean stochastic disturbances, but those results do not apply to limited predictions or non-i.i.d. disturbances. Other prior works analyze the power of

predictions in online optimization [11, 24], but the connection to online control in dynamical systems is unclear.

From this literature, fundamental questions about online control with predictions have emerged:

1. *What are the cost-optimal and regret-minimizing policies when given $k$ predictions? What are the corresponding cost and regret of these policies?*

2. *What is the marginal benefit from each additional prediction used by the policy, and how many predictions are needed to achieve (near-)optimal performance?*

3. *How well does MPC with $k$ predictions perform compared to cost-optimal and regret-minimizing policies?*

**Main contributions.**    We systematically address each of the questions above in the context of LQR systems with general stochastic and adversarial disturbances in the dynamics. In the stochastic case, we explicitly derive the cost-optimal and dynamic regret minimizing policies with $k$ predictions. In both the stochastic and adversarial cases, we derive (mostly tight) upper bounds for the optimal cost and minimum dynamic regret given access to $k$ predictions. We also show that the marginal benefit of an extra prediction exponentially decays as $k$ increases. Additionally, for MPC specifically, we show that it has a bounded performance ratio against the cost-optimal policy in both stochastic and adversarial settings. We further show that MPC is near-optimal in terms of dynamic regret, and needs only $O(\log T)$ predictions to achieve $O(1)$ dynamic regret (the same order as is needed by the dynamic regret minimizing policy) in both settings.

We would like to emphasize the generality of the results. The model we consider is the general LQR setting with disturbance in the dynamics, where only the stabilizability of $[A, B]$ and $[A^\top, Q]$ is assumed [4]. Further, in the stochastic setting we consider general distributions, which are not necessarily i.i.d. or zero-mean. Additionally, our results compare to the *globally* optimal policies for cost and regret rather than compare to the optimal linear or static policy. Finally, our upper bounds are (almost) *tight*, i.e., there exist some systems such that the bounds are (nearly) reached, up to lower-order terms.

It is perhaps surprising that classic MPC, which is a simple greedy policy (up to the prediction horizon), is near-optimal even with adversarial disturbances in the dynamics. Our results thus highlight the power of predictions to reduce the need for algorithmic sophistication. In that sense, our results somewhat mirror recent developments in the study of exploration strategies in online LQR control with unknown dynamics $\{A, B\}$: after a decade's research beginning with the work of Abbasi-Yadkori and Szepesvári [1], Simchowitz and Foster [29] recently show that naive exploration is optimal. Taken together with the result from [29], our paper provides additional evidence for the idea that the structure of LQR allows simple algorithmic ideas to be effective, which sheds light on key algorithmic principles and fundamental limits in continuous control.

## 2   Background and model

We consider the *Linear Quadratic Regulator (LQR)* optimal control problem with disturbances in the dynamics. In particular, we consider a linear system initialized with $x_0 \in \mathbb{R}^n$ and controlled by $u_t \in \mathbb{R}^d$, with dynamics

$$x_{t+1} = Ax_t + Bu_t + w_t \quad \text{and cost} \quad J = \sum_{t=0}^{T-1} (x_t^\top Q x_t + u_t^\top R u_t) + x_T^\top Q_f x_T,$$

where $T \geq 1$ is the total length of the control period. The goal of the controller is to minimize the cost given $A, B, Q, R, Q_f, x_0$, and the characterization of the disturbance $w_t$. Throughout this paper, we use $\rho(\cdot)$ to denote the spectral radius of a matrix and $\|\cdot\|$ to denote the 2-norm of a vector or the spectral norm of a matrix.

We assume $Q, Q_f \succeq 0, R \succ 0$ and the pair $(A, B)$ is *stabilizable*, i.e., there exists a matrix $K_0 \in \mathbb{R}^{d \times n}$ such that $\rho(A - BK_0) < 1$. Further, we assume the pair $(A, Q)$ is *detectable*, i.e., $(A^\top, Q)$ is stabilizable, to guarantee stability of the closed-loop. Note that detectability of $(A, Q)$ is more general than $Q \succ 0$, i.e., $Q \succ 0$ implies $(A, Q)$ is detectable. For $w_t$, in the stochastic case,

we assume $\{w_t\}_{t=0,1,\dots}$ are sampled from a joint distribution with bounded cross-correlation, i.e., $\mathbb{E}\left[w_t^\top w_{t'}\right] \leq m$ for any $t, t'$; in the adversarial case, we assume $w_t$ is picked from a bounded set $\Omega$.

These are standard assumptions in the literature, e.g., [13, 15, 29] and it is worth noting that our notion of stochasticity is much more general than typically considered [10, 12, 13]. We also note that many important problems can be straightforwardly converted to our model — for example, input-disturbed systems and the Linear Quadratic (LQ) tracking problem [4].

**Example: linear quadratic tracking.** The standard quadratic tracking problem is defined with dynamics $x_{t+1} = Ax_t + Bu_t + \tilde{w}_t$ and cost function $J = \sum_{t=0}^{T-1}(x_{t+1} - d_{t+1})^\top Q(x_{t+1} - d_{t+1}) + u_t^\top Ru_t$, where $\{d_t\}_{t=1}^T$ is the desired trajectory to track. To map this to our model, let $\tilde{x}_t = x_t - d_t$. Then, we get $J = \sum_{t=0}^{T-1}\tilde{x}_{t+1}^\top Q\tilde{x}_{t+1} + u_t^\top Ru_t$ and $\tilde{x}_{t+1} = A\tilde{x}_t + Bu_t + w_t$, which is an LQR control problem with disturbance $w_t = \tilde{w}_t + Ad_t - d_{t+1}$ in the dynamics.

## 2.1 Predictions

In the classic model, at each step $t$, the controller decides $u_t$ after observing $w_{t-1}$ and $x_t$. In other words, $u_t$ is a function of all the previous information: $x_0, x_1, \dots, x_{t-1}$ and $w_0, w_1, \dots, w_{t-1}$, or equivalently, of $x_0, w_0, w_1, \cdots, w_{t-1}$. We describe this scenario via the following *event sequence*:

$$x_0 \quad u_0 \quad w_0 \quad u_1 \quad w_1 \quad \cdots \quad u_{T-1} \quad w_{T-1},$$

where each $u_t$ denotes the decision of a control policy, each $w_t$ denote the observation of a disturbance, and each decision may depend on previous events.

However, in many real-world applications the controller may have some knowledge about future. In particular, at time step $t$, the controller may have *predictions* of immediate $k$ future disturbances and make decision $u_t$ based on $x_0, w_0, \dots, w_{t+k-1}$. In this case, the event sequence is given by:

$$x_0 \quad w_0 \quad w_1 \quad \cdots \quad w_{k-1} \quad u_0 \quad w_k \quad u_1 \quad w_{k+1} \quad \cdots \quad u_{T-k-1} \quad w_{T-1} \quad u_{T-k} \quad \cdots \quad u_{T-1}.$$

The existence of predictions is common in many applications such as disturbance estimation in robotics [27] and model predictive control (MPC) [9], which is a common approach for the LQ tracking problem. When given $k$ predictions of $d_t$, the LQ tracking problem can be formulated as a LQR problem with $k-1$ predictions of future disturbances. In this paper we assume all the predictions are *exact*, and leave inexact predictions [11, 28] as future work. This is common in the literature on online algorithms with predictions, e.g., [23, 24].

## 2.2 Disturbances

The characteristics of the disturbances have a fundamental impact on the optimal control policy and cost. We consider two types of disturbance: stochastic disturbances, which are drawn from a joint distribution (not necessarily i.i.d.), and adversarial disturbances, which are chosen by an adversary to maximize the overall control cost of the policy.

In the stochastic setting, we model the disturbance sequence $\{w_t\}_{t=0}^{T-1}$ as a discrete-time stochastic process with joint distribution $\mathcal{W}$ which is known to the controller. Let $W_t = W_t(w_0, \dots, w_{t-1})$ be the conditional distribution of $w_t$ given $w_0, \dots, w_{t-1}$. Then the cost of the optimal online policy with $k$ predictions is given by:

$$STO_k^T = \mathop{\mathbb{E}}_{w_0 \sim W_0, \dots, w_{k-1} \sim W_{k-1}}\left(\min_{u_0}\left(\mathop{\mathbb{E}}_{w_k \sim W_k}\left(\cdots \min_{u_{T-k-1}}\left(\mathop{\mathbb{E}}_{w_{T-1} \sim W_{T-1}}\left(\min_{u_{T-k}, \dots, u_{T-1}} J\right)\right)\right)\right)\right).$$

Note that the cost $J = J(x_0, u_0, \cdots, u_{T-1}, w_0, \cdots, w_{T-1})$. Two extreme cases are noteworthy: $k = 0$ reduces to the classic case without prediction and $k = T$ reduces to the offline optimal.

In the adversarial setting, each disturbance $w_t$ is selected by an adversary from a bounded set $\Omega \subseteq \mathbb{R}^n$ in order to maximize the cost. The controller has no information about the disturbance except that it is in $\Omega$. Similar to the stochastic setting, we define:

$$ADV_k^T = \sup_{w_0, \dots, w_{k-1} \in \Omega}\left(\min_{u_0}\left(\sup_{w_k \in \Omega}\left(\cdots \min_{u_{T-k-1}}\left(\sup_{w_{T-1} \in \Omega}\left(\min_{u_{T-k}, \dots, u_{T-1}} J\right)\right)\right)\right)\right).$$

This can be viewed as online $\mathcal{H}_\infty$ control [31] with predictions.

---
**Algorithm 1:** Model predictive control with $k$ predictions
---
**Parameter:** $\{A, B, Q, R\}$ and $\tilde{Q}_f \in \mathbb{R}^{n \times n}$
**Input:** $x_0, w_0, \ldots, w_{k-1}$
**1 for** $t = 0$ **to** $T - 1$ **do**
    **Input:** $x_t, w_{t+k-1}$      // The controller now knows $x_0, \ldots, x_t, w_0, \ldots, w_{t+k-1}$
**2**     $(u_t, \ldots, u_{t+k-1}) = \arg\min_u \sum_{i=t}^{t+k-1}(x_i^\top Q x_i + u_i^\top R u_i) + x_{t+k}^\top \tilde{Q}_f x_{t+k}$ subject to
      $x_{i+1} = A x_i + B u_i + w_i$ for $i = t, \ldots, t + k - 1$
    **Output:** $u_t$
---

The average cost in an infinite horizon is particularly important in both control and learning communities to understand asymptotic behaviors. We use separate notation for it:

$$\mathsf{STO}_k = \lim_{T \to \infty} \frac{1}{T} STO_k^T, \quad \mathsf{ADV}_k = \lim_{T \to \infty} \frac{1}{T} ADV_k^T.$$

We emphasize that we do not have any constraints (like linearity) on the policy space, and both $STO_k^T$ and $ADV_k^T$ are globally optimal with the corresponding type of disturbance. This point is important in light of recent results that show that linear policies cannot make use of predictions at all [16, 28], i.e., the cost of the best linear policy with infinite predictions ($k = \infty$) is asymptotically equal to that with no predictions ($k = 0$) in the setting with i.i.d. zero-mean stochastic disturbances. In this paper, we explicitly derive the optimal policy for every $k > 0$, which is *nonlinear* in general.

### 2.3 Model predictive control

Model predictive control (MPC) is perhaps the most common control policy for situations where predictions are available. MPC is a greedy algorithm with a receding horizon based on all available current predictions. Algorithm 1 provides a formal definition, and we additionally refer the reader to the book [9] for a literature review on MPC. We adopt a conventional definition of MPC as an online optimal control problem with a finite-time horizon with dynamics constraints. Note that other prior work on MPC sometimes considers other input and state constraints [9].

MPC is a practical algorithm in many scenarios like robotics [8], energy system [30] and data center cooling [22]. The existing theoretical studies of MPC focus on asymptotic stability and performance [6, 7, 18, 19, 25]. To our knowledge, we provide the first general, dynamic regret guarantee for MPC in this paper.

In this paper, we study the performance of MPC in three different cases, where disturbances are i.i.d. zero-mean stochastic, generally stochastic, and adversarial, corresponding to Sections 3 to 5 respectively. We define the performance of MPC in the stochastic and adversarial settings as follows:

$$MPCS_k^T = \mathop{\mathbb{E}}_{w_0, \ldots, w_{T-1}} J^{\mathsf{MPC}_k}, \qquad \mathsf{MPCS}_k = \lim_{T \to \infty} \frac{1}{T} MPCS_k^T,$$

$$MPCA_k^T = \sup_{w_0, \ldots, w_{T-1}} J^{\mathsf{MPC}_k}, \qquad \mathsf{MPCA}_k = \lim_{T \to \infty} \frac{1}{T} MPCA_k^T,$$

where $J^{\mathsf{MPC}_k}$ is the cost of MPC given a specific disturbance sequence, i.e., $J^{\mathsf{MPC}_k}(w) = J(u, w)$ where for each $t$, $u_t = \phi(x_t, w_t, \ldots, w_{t+k-1})$ and $\phi(\cdot)$ is the function that maps $x_t, w_t, \ldots, w_{t+k-1}$ to the policy $u_t$, as defined in Algorithm 1. By definition, $\mathsf{MPCS}_k \geq \mathsf{STO}_k$ and $\mathsf{MPCA}_k \geq \mathsf{ADV}_k$ for every $k \geq 1$ since they use the same information but the latter ones are defined to be optimal.

### 2.4 Dynamic regret and the performance ratio

In this paper, we focus on two performance metrics, the *dynamic regret* and the *performance ratio*.

**Dynamic regret.** Regret is a standard metric in online learning and provides a bound on the cost difference between an online algorithm and the optimal static policy given complete information. We focus on the *dynamic* regret, which compares to the optimal dynamic offline policy, rather than the optimal static offline policy. Note that the optimal offline policy may be nonlinear. It is important to

consider nonlinear policies because recent results highlight that the optimal offline policy can have cost that is arbitrarily lower than the optimal linear policy in hindsight [16, 28].

More specifically, we compare the cost of an online algorithm with $k$ predictions to that of the offline optimal (nonlinear) algorithm, i.e., one that has predictions of all disturbances. For MPC with $k$ predictions, we define its dynamic regret in the stochastic and adversarial settings, respectively, as:

$$Reg^S(\mathsf{MPC}_k) = \mathop{\mathbb{E}}_{(w_0,\cdots,w_{T-1})\sim\mathcal{W}} \left( J^{\mathsf{MPC}_k}(w) - \min_{u'_0,\dots,u'_{T-1}} J(u',w) \right),$$

$$Reg^A(\mathsf{MPC}_k) = \sup_{w_0,\cdots,w_{T-1}\in\Omega} \left( J^{\mathsf{MPC}_k}(w) - \min_{u'_0,\dots,u'_{T-1}} J(u',w) \right).$$

As compared to (static) regret, dynamic regret does not have any restriction on the policies $u'_0,\dots,u'_{T-1}$ used for comparison and thus differs from other notions of regret where $u'_0,\dots,u'_{T-1}$ are limited in special cases. For example, in the classic form of regret, $u'_0 = \cdots = u'_{T-1}$; and in the regret compared to the best offline linear controller [2, 12], $u'_t = -K^*x_t$.

In this work, we obtain both upper bounds and lower bounds on dynamic regret. For lower bounds, we define the minimum possible regret that an algorithm with $k$ predictions can achieve (i.e., the regret of the algorithm that minimizes the regret):

$$Reg_k^{S^*} = \mathop{\mathbb{E}}_{w_0,\cdots,w_{k-1}} \min_{u_0} \mathop{\mathbb{E}}_{w_k} \cdots \min_{u_{T-k-1}} \mathop{\mathbb{E}}_{w_{T-1}} \min_{u_{T-k},\cdots,u_{T-1}} \left( J(u,w) - \min_{u'_0,\dots,u'_{T-1}} J(u',w) \right),$$

$$Reg_k^{A^*} = \sup_{w_0,\cdots,w_{k-1}} \min_{u_0} \sup_{w_k} \cdots \min_{u_{T-k-1}} \sup_{w_{T-1}} \min_{u_{T-k},\cdots,u_{T-1}} \left( J(u,w) - \min_{u'_0,\dots,u'_{T-1}} J(u',w) \right).$$

Finally, we end our discussion of dynamic regret with a note highlighting an important contrast between stochastic and adversarial settings. In the stochastic setting,

$$Reg_k^{S^*} = \mathop{\mathbb{E}}_{w_0,\cdots,w_{k-1}} \min_{u_0} \mathop{\mathbb{E}}_{w_k} \cdots \min_{u_{T-k-1}} \mathop{\mathbb{E}}_{w_{T-1}} \left( \min_{u_{T-k},\cdots,u_{T-1}} J(u,w) - \min_{u'_0,\dots,u'_{T-1}} J(u',w) \right)$$

$$= \mathop{\mathbb{E}}_{w_0,\cdots,w_{k-1}} \min_{u_0} \mathop{\mathbb{E}}_{w_k} \cdots \min_{u_{T-k-1}} \mathop{\mathbb{E}}_{w_{T-1}} \min_{u_{T-k},\cdots,u_{T-1}} J(u,w) - \mathop{\mathbb{E}}_{w_0,\dots,w_{T-1}} \min_{u'_0,\dots,u'_{T-1}} J(u',w)$$

$$= STO_k^T - STO_T^T.$$

This equality still holds if we take $\arg\min$ instead of $\min$ and thus the regret-optimal policy is the same as the cost-optimal policy. However, in the adversarial case, a similar reasoning gives an inequality: $Reg_k^{A^*} \geq ADV_k^T - ADV_T^T$, and correspondingly, the regret-optimal and cost-optimal policies can be different. Similarly, for MPC, we have $Reg^S(\mathsf{MPC}_k) = MPCS_k^T - STO_T^T$ while $Reg^A(\mathsf{MPC}_k) \geq MPCA_k^T - ADV_T^T$.

**Performance ratio.** The second metric we study is a new metric that we term the *performance ratio*. It characterizes the ratio of the cost of an online algorithm with $k$ predictions to the cost of the optimal online algorithm using $k$ predictions. Thus, it gives a way of comparing to a weaker benchmark than regret – one that has the same amount of information as the algorithm. Note that it is related to, but different than, the competitive ratio in this context. Formally, the performance ratio of the MPC algorithm in stochastic and adversarial settings, respectively, is defined as:

$$PR^S(\mathsf{MPC}_k) = \frac{\mathsf{MPCS}_k}{\mathsf{STO}_k}, \quad PR^A(\mathsf{MPC}_k) = \frac{\mathsf{MPCA}_k}{\mathsf{ADV}_k}.$$

While the dynamic regret indicates whether the algorithm can match the optimal offline policy (which has complete information), the performance ratio measures whether the algorithm is using the information available to it in as efficient a manner as possible. Thus, the contrast between the two separates the efficiency of the algorithm from the inefficiency created by the lack of information about future disturbances.

Finally, one may wonder if there are connections between dynamic regret and performance ratio. As might be expected, in both the stochastic and adversarial settings, the performance ratio of an online policy with $k$ predictions provides a lower bound of its dynamic regret:

$$PR^S(\mathsf{MPC}_k) - 1 \leq \frac{\mathsf{MPCS}_k}{\mathsf{STO}_\infty} - 1 = \frac{\mathsf{MPCS}_k - \mathsf{STO}_\infty}{\mathsf{STO}_\infty} = \frac{1}{\mathsf{STO}_\infty} \lim_{T\to\infty} \frac{1}{T} Reg^S(\mathsf{MPC}_k),$$

$$PR^A(\mathsf{MPC}_k) - 1 \leq \frac{\mathsf{MPCA}_k}{\mathsf{ADV}_\infty} - 1 = \frac{\mathsf{MPCA}_k - \mathsf{ADV}_\infty}{\mathsf{ADV}_\infty} \leq \frac{1}{\mathsf{ADV}_\infty} \lim_{T \to \infty} \frac{1}{T} Reg^A(\mathsf{MPC}_k).$$

## 3  Zero-mean i.i.d. disturbances

We begin our analysis with the simplest of the three settings we consider: the disturbances $w_t$ are independent and identically distributed with zero mean. Though i.i.d. zero-mean is a limited setting, it is still complex enough to study predictions and the first results characterizing the optimal policy with predictions appeared only recently [15, 16], focusing only on the optimal policy when $k \to \infty$.

Before delving into our results, we first recap the classic *Infinite Horizon Linear Quadratic Stochastic Regulator* [4, 5], i.e., the case when $k = 0$:

**Proposition 3.1** (Anderson and Moore [5])**.** *Let $w_t$ be i.i.d. with zero mean and covariance matrix $W$. Then, the optimal control policy corresponding to $\mathsf{STO}_0$ is given by:*

$$u_t = -(R + B^\top PB)^{-1}B^\top PAx_t =: -Kx_t,$$

*where $P$ is the solution of discrete-time algebraic Riccati equation (DARE)*

$$P = Q + A^\top PA - A^\top PB(R + B^\top PB)^{-1}B^\top PA. \tag{1}$$

*The corresponding closed-loop dynamics $A - BK$ is exponentially stable, i.e., $\rho(A - BK) < 1$. Further, the optimal cost is given by $\mathsf{STO}_0 = \mathrm{Tr}\{PW\}$.*

This result has been extensively studied in optimal control theory [4, 21] as well as in reinforcement learning [13, 14, 29]. We want to emphasize two important properties of the optimal policy $u_t = -Kx_t$. First, the policy is *linear* in the state $x_t$. In contrast, we show later that the optimal policy when $k \neq 0$ is, in general, *nonlinear*. Second, under the assumptions of our model, this policy is *exponentially stable*, i.e., $\rho(A - BK) < 1$. We leverage this to show the power of predictions later in the paper.

**Optimal policy.**   Let $F = A - BK$ and $\lambda = \frac{1+\rho(F)}{2} < 1$. From Gelfand's formula, there exists a constant $c(n)$ such that $\|F^k\| \leq c(n)\lambda^k$ for all $k \geq 1$.

**Theorem 3.2.** *Let $w_t$ be i.i.d. with zero mean and covariance matrix $W$. Suppose the controller has $k \geq 1$ predictions. Then, the optimal control policy at each step $t$ is given by:*

$$u_t = -(R + B^\top PB)^{-1}B^\top \left( PAx_t + \sum_{i=0}^{k-1}(A^\top - A^\top PH)^i Pw_{t+i} \right), \tag{2}$$

*where $P$ is the solution of DARE in Equation* (1)*. The cost under this policy is:*

$$\mathsf{STO}_k = \mathrm{Tr}\left\{ \left( P - \sum_{i=0}^{k-1} P(A - HPA)^i H(A^\top - A^\top PH)^i P \right) W \right\}, \tag{3}$$

*where $H = B(R + B^\top PB)^{-1}B^\top$.*

Following the approach developed in [15, 16], the proof is based on an analysis of quadratic cost-to-go functions in the form $V_t(x_t) = x_t^\top P_t x_t + v_t^\top x_t + q_t$. Note that $A - HPA = A - B(R + B^\top PB)^{-1}B^\top PA = A - BK = F$. Thus, the online optimal cost $\mathsf{STO}_k$ with $k$ predictions approaches the offline optimal cost $\mathsf{STO}_\infty$ by an exponential rate. In other words, $\mathsf{STO}_k/\mathsf{STO}_\infty = 1 + O(\|F^k\|^2) = 1 + O(\lambda^{2k})$. Two extreme cases of our result are noteworthy. When $k = 0$, it reduces to the classic Proposition 3.1. When $k \to \infty$, it reduces to the offline optimal case derived by Goel and Hassibi [16].

**Model predictive control.**   As might be expected, since the disturbances are i.i.d., future disturbances have no dependence on the current. As a result, MPC gives the *optimal* policy.

**Theorem 3.3.** *In Algorithm 1, let $\tilde{Q}_f = P$. Then, the MPC policy with $k$ predictions is also given by Equation* (2)*. Assuming i.i.d. disturbance with zero mean, the MPC policy is optimal.*

Due to the greedy nature, MPC does not utilize any properties of the disturbance, so the first part in Theorem 3.3 holds not only for i.i.d. disturbance, but also other types of disturbance considered in the later sections, i.e., MPC policy with $k$ predictions is always given by Equation (2).

# 4 General stochastic disturbances

In this section, we consider a general form of stochastic disturbance, more general than typically considered in this context [10, 12, 13]. Suppose the disturbance sequence $\{w_t\}_{t=0,1,2,\dots}$ is sampled from a joint distribution $\mathcal{W}$ such that the trace of the cross-correlation of each pair is uniformly bounded, i.e., there exist $m > 0$ such that for all $t, t' \geq 1$, $\mathbb{E}\big[w_t^\top w_{t'}\big] \leq m$.

**Optimal policy.** In the case of general stochastic disturbances, we cannot obtain as clean a form for $\mathsf{STO}_k$ as in the i.i.d. case in Section 3. However, the marginal benefit of having an extra prediction decays with the same (exponential) rate and the optimal policy is similar to that in Section 3, but with some additional terms that characterize the expected future disturbances given the current information.

**Theorem 4.1.** *The optimal control policy with general stochastic disturbance is given by:*

$$u_t = -(R + B^\top PB)^{-1}B^\top \left( PAx_t + \sum_{i=0}^{k-1} F^{\top i} P w_{t+i} + \sum_{i=k}^{\infty} F^{\top i} P \mu_{t+i|t+k-1} \right), \qquad (4)$$

*where $\mu_{t'|t} = \mathbb{E}[w_{t'} \,|\, w_0, \dots, w_t]$. Under this policy, the marginal benefit of obtaining an extra prediction decays exponentially fast in the existing number $k$ of predictions. Formally, for $k \geq 1$,*

$$\mathsf{STO}_k - \mathsf{STO}_{k+1} = O(\|F^k\|^2) = O(\lambda^{2k}).^1$$

This proof leverages a novel difference analysis of cost-to-go functions. Note that for some distributions, $\mathsf{STO}_k$ may approach $\mathsf{STO}_\infty$ much faster than exponential rate. It is even possible that $\mathsf{STO}_k = \mathsf{STO}_\infty$ for finite $k$, as we show in Example 4.2 below. On the other hand, there are scenarios where $\mathsf{STO}_k$ approaches $\mathsf{STO}_\infty$ in an exactly exponential manner, as we show in Example 4.3 below.

**Example 4.2.** *Define the joint distribution $\mathcal{W}$ such that with probability $1/2$, all $w_t = w$, and otherwise all $w_t = -w$. In this case, one prediction is equivalent to infinite predictions since it is enough to distinguish these two scenarios with only $w_0$. As a result, $\mathsf{STO}_1 = \mathsf{STO}_\infty$.*

**Example 4.3.** *Suppose the system is 1-d ($n = d = 1$) and the disturbance is i.i.d. with zero mean, i.e., the setting of Section 3. Then, according to Equation (3), as long as $F, P, H, W$ are non-zero,*

$$\mathsf{STO}_k - \mathsf{STO}_\infty = \sum_{i=k}^{\infty} F^{2i} P^2 H W = \Theta(F^{2k}).$$

**Model predictive control.** The comparison between the MPC policy in Equation (2) and the optimal policy in Equation (4) reveals that MPC is a *truncation* of the optimal policy and is no longer optimal because MPC is a greedy policy without considering future dependence on current information. Nevertheless, it is still a near-optimal policy, as characterized by the following results.

**Theorem 4.4.** $\mathsf{MPCS}_k - \mathsf{MPCS}_{k+1} = O(\|F^k\|^2) = O(\lambda^{2k})$. *Moreover, in Example 4.3, $\mathsf{MPCS}_k - \mathsf{MPCS}_{k+1} = \Theta(\|F^k\|^2)$.*

In other words, the marginal benefit for the MPC algorithm of an extra prediction decays exponentially fast, paralleling the result for optimal policy in Equation (4). Theorem 4.4 implies that MPC has a bounded performance ratio, which converges to 1 with an exponential rate in the number of available predictions. Formally:

**Corollary 4.5.** $PR^S(\mathsf{MPC}_k) = \frac{\mathsf{MPCS}_k}{\mathsf{STO}_k} \leq \frac{\mathsf{MPCS}_k}{\mathsf{STO}_\infty} = \frac{\mathsf{MPCS}_k}{\mathsf{MPCS}_\infty} = 1 + O(\|F^k\|^2) = 1 + O(\lambda^{2k})$. *Moreover, in Example 4.2, we have $PR^S(\mathsf{MPC}_k) = 1 + \Theta(\|F^k\|^2)$.*

Besides, the dynamic regret of MPC (nearly) matches the order of the optimal dynamic regret.

**Theorem 4.6** (Main result)**.** $Reg^S(\mathsf{MPC}_k) = MPCS_k^T - STO_T^T = O(\|F^k\|^2 T + 1) = O(\lambda^{2k} T + 1)$, *where the second term results from the difference between finite/infinite horizons.*

**Theorem 4.7.** *The optimal dynamic regret* $Reg_k^{S^*} = STO_k^T - STO_T^T = O(\|F^k\|^2 T + 1) = O(\lambda^{2k}T + 1)$ *and there exist* $A$, $B$, $Q$, $R$, $Q_f$, $x_0$, *and* $\mathcal{W}$ *such that* $Reg_k^{S^*} = \Theta(\|F^k\|^2(T - k))$.

Note that, in the stochastic case, the regret-optimal policy is the same as the cost-optimal policy, i.e., the policy for $STO_k^T$ is the same as $Reg_k^{S^*}$.

## 5   Adversarial disturbances

We now move from stochastic to adversarial disturbances. In this section, the disturbances are chosen from a bounded set $\Omega \subseteq \mathbb{R}^n$ by an adversary in order to maximize the controller's cost. Maintaining small regret is more challenging in adversarial models than in stochastic ones, so one may expect weaker bounds. Perhaps surprisingly, we obtain bounds with the same order.

**Optimal policy.**   In the adversarial setting, the cost of the optimal policy, defined with a sequence of $\min$'s and $\sup$'s, is the equilibrium value of a two-player zero-sum game. In general, it is impossible to give an analytical expression of either $\mathsf{ADV}_k$ or the corresponding optimal policy. However, we prove a result that is structurally similar to the results from the stochastic setting, highlighting the exponential improvement from predictions.

**Theorem 5.1.** *For* $k \geq 1$, $\mathsf{ADV}_k - \mathsf{ADV}_{k+1} = O(\|F^k\|^2) = O(\lambda^{2k})$.

Similarly to Example 4.2 for the stochastic case, in the adversarial setting, the optimal cost with $k$ predictions may approach the offline optimal cost (under infinite predictions) much faster than exponential rate, and it is possible that $\mathsf{ADV}_k = \mathsf{ADV}_\infty$ for finite $k$, as shown in Example 5.2.

**Example 5.2.** *Let* $A = B = Q = R = 1$ *and* $\Omega = [-1, 1]$. *In this case, one prediction is enough to leverage the full power of prediction. Formally, we have* $\mathsf{ADV}_1 = \mathsf{ADV}_\infty = 1$. *In other words, for all* $k \geq 1$, $\mathsf{ADV}_k = 1$. *The optimal control policy (as* $T \to \infty$) *is a piecewise function:*

$$u^*(x, w) = \begin{cases} -(x + w) & , -1 \leq x + w \leq 1 \\ -(x + w) + \frac{3 - \sqrt{5}}{2}(x + w - 1) & , x + w > 1 \\ -(x + w) + \frac{3 - \sqrt{5}}{2}(x + w + 1) & , x + w < -1 \end{cases}.$$

*The proof leverages two different cost-to-go functions for the* $\min$ *player and the* $\sup$ *player.*

Note that the optimal policy could be much more complex. Unlike Example 5.2, where the optimal policy is piecewise linear with only 3 pieces, for other values of $A, B, Q, R$, this function may have many more pieces.

**Model predictive control.**   Under adversarial disturbances, MPC is suboptimal, e.g., in Example 5.2. However, its performance ratio and dynamic regret bounds turn out to be the same as those in the stochastic setting.

**Theorem 5.3.** $\mathsf{MPCA}_k - \mathsf{MPCA}_{k+1} = O(\|F^k\|^2) = O(\lambda^{2k})$.

**Corollary 5.4.** *For* $k \geq 1$, $PR^A(\mathsf{MPC}_k) = \frac{\mathsf{MPCA}_k}{\mathsf{ADV}_k} \leq \frac{\mathsf{MPCA}_k}{\mathsf{ADV}_\infty} = \frac{\mathsf{MPCA}_k}{\mathsf{MPCA}_\infty} = 1 + O(\|F^k\|^2) = 1 + O(\lambda^{2k})$.

This highlights that MPC has a bounded performance ratio, which converges to 1 with exponential rate. Additionally, MPC has the same order of dynamic regret as the stochastic case:

**Theorem 5.5** (Main result). $Reg^A(\mathsf{MPC}_k) = O(\|F^k\|^2 T + 1) = O(\lambda^{2k}T + 1)$.

This dynamic regret is linear in the horizon $T$ if we fix the number of predictions. However, if $k$ is a super-constant function of $T$ — an increasing function of $T$ that is not upper-bounded by a constant — then the regret is sub-linear. Furthermore, if we let $k = \frac{\log T}{2 \log(1/\lambda)}$, then $Reg^A(\mathsf{MPC}_k) = O(1)$. In other words, we can get constant regret with $O(\log T)$ predictions, even with adversarial disturbances. Finally, as implied by the following result, the $O(\log T)$ horizon cannot be improved since even the regret minimizing algorithm needs the same order of predictions to reach constant regret.

**Theorem 5.6.** $Reg_k^{A^*} = O(\|F^k\|^2 T + 1) = O(\lambda^{2k}T + 1)$. *Moreover, there exist* $A$, $B$, $Q$, $R$, $Q_f$, $x_0$, *and* $\Omega$ *such that* $Reg_k^{A^*} = \Omega(\|F^k\|^2(T - k))$.[2]

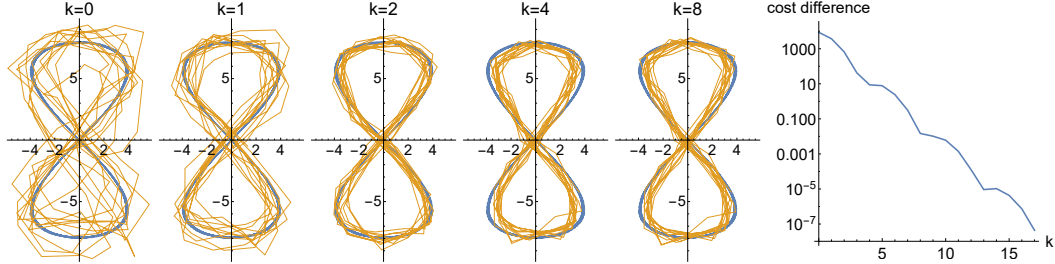

Figure 1: The power of predictions in online tracking. The left four figures show the desired trajectory (blue) and the actual trajectories (orange). The rightmost figure shows the cost difference (regret) between MPC using $k$ predictions and the offline optimal policy. Note that the y-axis of the rightmost figure is in log-scale.

## 6   Numerical experiments

To illustrate our theoretical results, we test MPC with different numbers of predictions in a Linear Quadratic (LQ) tracking problem, where the desired trajectory is given by:

$$d_t = \begin{bmatrix} 8\sin(t/3)\cos(t/3) \\ 8\sin(t/3) \end{bmatrix}.$$

We consider following double integrator dynamics:

$$p_{t+1} = p_t + v_t + h_t, \qquad v_{t+1} = v_t + u_t + \eta_t,$$

where $p_t \in \mathbb{R}^2$ is the position, $v_t$ is the velocity, $u_t$ is the control, and $h_t, \eta_t \sim \mathrm{U}[-1,1]^2$ are i.i.d. noises. The objective is to minimize

$$\sum_{t=0}^{T-1} \|p_t - d_t\|^2 + \|u_t\|^2,$$

where we let $T = 200$. This problem can be converted to the standard LQR with disturbance $w_t$ by letting $x_t = \begin{bmatrix} p_t \\ v_t \end{bmatrix}$ and $\tilde{w}_t = \begin{bmatrix} h_t \\ \eta_t \end{bmatrix}$ and then using the reduction in the LQ tracking example in Section 2. Note that after the reduction, the disturbances are the combination of a deterministic trajectory and i.i.d. noises, which corresponds to the case discussed in Section 4.

Figure 1 shows the tracking results with MPC using different numbers of predictions. We see that the regret exponentially decreases as the number of predictions increases, which is consistent with our theoretical results.

## 7   Concluding remarks

We conclude with several open problems and potential future research directions. Our results highlight the power of predictions and show that, given predictions, a simple greedy policy (MPC) is near-optimal for LQR control with disturbances in the dynamics, in terms of dynamic regret. Building on our results, it will be interesting to understand if MPC has a constant competitive ratio in this setting. In a different but related setting, Chen et al. [11] show for negative results on the competitive ratio so the answer is unclear at this point. Additionally, in this paper predictions are assumed to be perfect. Of course, in real applications predictions are noisy and are derived based on historical data. An important extension will be to understand how the analysis and results in this paper can extend to models with imperfect predictions learned from history, such as done in related models [11, 28]. Finally, real-world MPC problems often require non-linear dynamics and/or constraints, and the learning-theoretic study of such settings remains largely unexplored.

## Broader Impact

Linear quadratic control is a common and powerful model with a variety of commercial and industrial applications, e.g., in robotics, chemical process control, and energy systems. This paper provides new

fundamental insights about the role of predictions in online linear quadratic control with disturbances and provides the first finite time performance guarantees for the most commonly used policy in the linear quadratic setting, model predictive control (MPC).

The guarantees provided by the theoretical analysis in this paper offer the potential for ensuring safety and robustness in industry applications where predictions are common and MPC is used. However, like many other theoretical contributions, this paper's results are limited to its assumptions, e.g., linear system and fixed system parameters $\{A, B, Q, R\}$. The performance of MPC and the fundamental limits in other scenarios, e.g., nonlinear dynamics or time-variant $\{A, B, Q, R\}$, are still open research problems.

We see no ethical concerns related to the results in this paper.

## Acknowledgments and Disclosure of Funding

This project was supported in part by funding from Raytheon, DARPA PAI, AitF-1637598 and CNS-1518941, with additional support for Guanya Shi provided by the Simoudis Discovery Prize.

## Footnotes

[1]We say that $f(k) = O(g(k))$ if $\exists C > 0, \forall k \geq 1, |f(k)| \leq C\, g(k)$; $\Omega()$ is similar except that the last "$\leq$" is replaced by "$\geq$"; $\Theta()$ means both $O()$ and $\Omega()$. This is stronger than the standard definition where $f(k) = O(g(k))$ if $\exists C > 0, k^* > 0, \forall k \geq k^*, |f(k)| \leq C\, g(k)$.

[2] $\Omega(\cdot)$ is the growth order notation and has nothing to do with the bounded set $\Omega$.

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
