[Supplementary Material]

# A Proofs of Section 3

In all proofs in this paper, for a sequence $x = (x_0, x_1, \ldots, x_n)$, we use $x_{a:b}$ to denote its consecutive subsequence $(x_a, x_{a+1}, \ldots, x_b)$.

## A.1 Proof of Theorem 3.2

*Let $w_t$ be i.i.d. with zero mean and covariance matrix $W$. Suppose the controller has $k \geq 1$ predictions. Then, the optimal control policy at each step $t$ is given by:*

$$u_t = -(R + B^\top P B)^{-1} B^\top \left( P A x_t + \sum_{i=0}^{k-1} (A^\top - A^\top P H)^i P w_{t+i} \right), \tag{2}$$

*where $P$ is the solution of DARE in Equation (1). The cost under this policy is:*

$$\mathsf{STO}_k = \mathrm{Tr}\left\{ \left( P - \sum_{i=0}^{k-1} P(A - HPA)^i H (A^\top - A^\top P H)^i P \right) W \right\}, \tag{3}$$

*where $H = B(R + B^\top P B)^{-1} B^\top$.*

*Proof.* Our proof technique closely follows that in Section 4.1 of [16]. To begin, note that the definition of $STO_k^T$ has a structure of repeating min's and $\mathbb{E}$'s. We use dynamic programming to compute the value iteratively. In particular, we apply backward induction to solve the optimal cost-to-go functions, from time step $T$ to the initial state. Given state $x_t$ and predictions $w_t, \ldots, w_{t+k-1}$, we define the cost-to-go function:

$$V_t(x_t; w_{t:t+k-1}) := \min_{u_t} \mathbb{E}_{w_{t+k}} \min_{u_{t+1}} \cdots \mathbb{E}_{w_{T-1}} \min_{u_{T-k}, \cdots, u_{T-1}} \sum_{i=t}^{T-1} (x_i^\top Q x_i + u_i^\top R u_i) + x_T^\top Q_f x_T \tag{5}$$

$$= x_t^\top Q x_t + \min_{u_t} \left( u_t^\top R u_t + \mathbb{E}_{w_{t+k}} [V_{t+1}(A x_t + B u_t + w_t; w_{t+1:t+k})] \right)$$

with $V_T(x_T; \ldots) = x_T^\top Q_f x_T$. Note that $\mathbb{E}_{w_{t+k}}$ has no effect for $t \geq T - k$. This function measures the expected overall control cost from a given state to the end, assuming the controller makes the optimal decision at each time.

We will show by backward induction that for every $t = 0, \ldots, T$, $V_t(x_t; w_{t:t+k-1}) = x_t^\top P_t x_t + v_t^\top x_t + q_t$, where $P_t, v_t, q_t$ are coefficients that may depend on $w_{t:t+k-1}$. This is clearly true for $t = T$. Suppose this is true at $t + 1$. Then,

$$V_t(x; w_{t:t+k-1})$$
$$= x^\top Q x + \min_u \left( u^\top R u + (A x + B u + w_t)^\top P_{t+1} (A x + B u + w_t) \right.$$
$$\left. + \mathbb{E}_{w_{t+k}} [v_{t+1}]^\top (A x + B u + w_t) + \mathbb{E}_{w_{t+k}} [q_{t+1}] \right)$$
$$= x^\top Q x + (A x + w_t)^\top P_{t+1} (A x + w_t) + \mathbb{E}_{w_{t+k}} [v_{t+1}]^\top (A x + w_t) + \mathbb{E}_{w_{t+k}} [q_{t+1}]$$
$$+ \min_u \left( u^\top (R + B^\top P_{t+1} B) u + u^\top B^\top \left( 2 P_{t+1} A x + 2 P_{t+1} w_t + \mathbb{E}_{w_{t+k}} [v_{t+1}] \right) \right).$$

The optimal $u$ is obtained by setting the derivative to be zero:

$$u^* = -(R + B^\top P_{t+1} B)^{-1} B^\top \left( P_{t+1} A x + P_{t+1} w_t + \frac{1}{2} \mathbb{E}_{w_{t+k}} [v_{t+1}] \right). \tag{6}$$

Let $H_t = B(R + B^\top P_{t+1} B)^{-1} B^\top$. Plugging $u^*$ back into $V_t$, we have

$$V_t(x; w_{t:t+k-1})$$
$$= x^\top Q x + (A x + w_t)^\top P_{t+1} (A x + w_t) + \mathbb{E}_{w_{t+k}} [v_{t+1}]^\top (A x + w_t) + \mathbb{E}_{w_{t+k}} [q_{t+1}]$$

$$- \left( P_{t+1} A x + P_{t+1} w_t + \frac{1}{2} \mathop{\mathbb{E}}_{w_{t+k}} [v_{t+1}] \right)^\top H_t \left( P_{t+1} A x + P_{t+1} w_t + \frac{1}{2} \mathop{\mathbb{E}}_{w_{t+k}} [v_{t+1}] \right)$$

$$= x^\top \left( Q + A^\top P_{t+1} A - A^\top P_{t+1} H_t P_{t+1} A \right) x$$

$$+ x^\top \left( (A^\top - A^\top P_{t+1} H_t) \mathop{\mathbb{E}}_{w_{t+k}} [v_{t+1}] + 2(A^\top - A^\top P_{t+1} H_t) P_{t+1} w_t \right)$$

$$+ w_t^\top (P_{t+1} - P_{t+1} H_t P_{t+1}) w_t + w_t^\top (I - P_{t+1} H_t) \mathop{\mathbb{E}}_{w_{t+k}} [v_{t+1}]$$

$$- \frac{1}{4} \mathop{\mathbb{E}}_{w_{t+k}} [v_{t+1}]^\top H_t \mathop{\mathbb{E}}_{w_{t+k}} [v_{t+1}] + \mathop{\mathbb{E}}_{w_{t+k}} [q_{t+1}].$$

Thus, the recursive formulae, which parallel [16], are given by:

$$P_t = Q + A^\top P_{t+1} A - A^\top P_{t+1} H_t P_{t+1} A, \tag{7a}$$

$$v_t = (A^\top - A^\top P_{t+1} H_t) \mathop{\mathbb{E}}_{w_{t+k}} [v_{t+1}] + 2(A^\top - A^\top P_{t+1} H_t) P_{t+1} w_t, \tag{7b}$$

$$q_t = w_t^\top (P_{t+1} - P_{t+1} H_t P_{t+1}) w_t + w_t^\top (I - P_{t+1} H_t) \mathop{\mathbb{E}}_{w_{t+k}} [v_{t+1}]$$

$$- \frac{1}{4} \mathop{\mathbb{E}}_{w_{t+k}} [v_{t+1}]^\top H_t \mathop{\mathbb{E}}_{w_{t+k}} [v_{t+1}] + \mathop{\mathbb{E}}_{w_{t+k}} [q_{t+1}]. \tag{7c}$$

As $T - t \to \infty$, $P_t$ and $H_t$ converge to $P$ and $H$ respectively, where $P$ is the solution of discrete-time algebraic Riccati equation (DARE) $P = Q + A^\top P A - A^\top P H P A$, and $H = B(R + B^\top P B)^{-1} B^\top$. Note that $v_T = 0$ and $q_T = 0$. Then,

$$v_t = 2 \sum_{i=0}^{k-1} (A^\top - A^\top P H)^{i+1} P w_{t+i}, \tag{8}$$

$$q_t = w_t^\top (P - P H P) w_t + w_t^\top (I - P H) \mathop{\mathbb{E}}_{w_{t+k}} [v_{t+1}] - \frac{1}{4} \mathop{\mathbb{E}}_{w_{t+k}} [v_{t+1}]^\top H \mathop{\mathbb{E}}_{w_{t+k}} [v_{t+1}] + \mathop{\mathbb{E}}_{w_{t+k}} [q_{t+1}], \tag{9}$$

$$\mathop{\mathbb{E}}_{w_{t+k}} [v_{t+1}] = 2 \sum_{i=1}^{k-1} (A^\top - A^\top P H)^i P w_{t+i}. \tag{10}$$

Taking the expectation of $q_t$ over all randomness, namely $w_0, w_1, w_2, \ldots$, we have

$$\mathbb{E}[q_t] = \mathrm{Tr}\{(P - P H P) W\} - \sum_{i=1}^{k-1} \mathrm{Tr}\{P(A - H P A)^i H (A^\top - A^\top P H)^i P W\} + \mathbb{E}[q_{t+1}]$$

$$= \mathrm{Tr}\left\{ \left( P - \sum_{i=0}^{k-1} P(A - H P A)^i H (A^\top - A^\top P H)^i P \right) W \right\} + \mathbb{E}[q_{t+1}], \tag{11}$$

where in the first equality we use $\mathbb{E}[w_t] = 0$ and the independence of the disturbances. Thus, as $T \to \infty$, in each time step, a constant cost is incurred and the average cost $\mathsf{STO}_k$ is exactly this value.

$$\mathsf{STO}_k = \lim_{T \to \infty} \frac{1}{T} STO_k^T = \lim_{T \to \infty} \frac{1}{T} \mathbb{E}[V_0(x_0; w_{0:k-1})] = \lim_{T \to \infty} \frac{1}{T} \mathbb{E}[q_0]$$

$$= \lim_{T \to \infty} \frac{1}{T} \sum_{t=0}^{T-1} \mathbb{E}[q_t] - \mathbb{E}[q_{t+1}] = \mathrm{Tr}\left\{ \left( P - \sum_{i=0}^{k-1} P(A - H P A)^i H (A^\top - A^\top P H)^i P \right) W \right\}.$$

The explicit form of the optimal control policy is obtained by combining Equations (6) and (10). $\quad\square$

## A.2  Proof of Theorem 3.3

*In Algorithm 1, let $\tilde{Q}_f = P$. Then, the MPC policy with $k$ predictions is also given by Equation (2). Assuming i.i.d. disturbance with zero mean, the MPC policy is optimal.*

*Proof.* Due to the greedy nature, MPC policy is given by the solution of a length-$k$ optimal control problem, given deterministic $w_t, \cdots, w_{t+k-1}$. In other words, we want to derive the optimal policy $(u_t, \ldots, u_{t+k-1})$ that minimizes

$$\sum_{i=t}^{t+k-1} (x_i^\top Q x_i + u_i^\top R u_i) + x_{t+k}^\top P x_{t+k},$$

where $x_{i+1} = A x_i + B u_i + w_i$, given $x_t, w_t, \ldots, w_{t+k-1}$. Define the cost-to-go function at time $i$ given $x_i, w_i, \ldots, w_{t+k-1}$:

$$V_i(x_i; w_{i:t+k-1}) = \min_{u_{i:t+k-1}} \sum_{j=i}^{t+k-1} (x_j^\top Q x_j + u_j^\top R u_j) + x_{t+k}^\top P x_{t+k}$$

$$= x_i^\top Q x_i + \min_{u_i}(u_i^\top R u_i + V_{i+1}(A x_i + B u_i + w_i; w_{i+1:t+k-1})).$$

Note that $V_{t+k}(x_{t+k}) = x_{t+k}^\top P x_{t+k}$. Similar to the proof of Theorem 3.2, we can inductively show that $V_i(x_i; w_{i:t+k-1}) = x_i^\top P x_i + v_i^\top x_i + q_i$ for some $v_i$ and $q_i$. Note that the second-degree coefficient no longer depends on the index $i$ as in the previous proof because we start from $P$, the solution of DARE. We then have the followings equations that parallel with Equations (6) and (8):

$$v_i = 2 \sum_{j=0}^{t+k-i-1} F^{\top j+1} P w_{i+j},$$

$$u_i^* = -(R + B^\top P B)^{-1} B^\top \left( P A x_i + P w_i + \frac{1}{2} v_{i+1} \right)$$

$$= -(R + B^\top P B)^{-1} B^\top \left( P A x_i + \sum_{j=0}^{t+k-i-1} F^{\top j} P w_{i+j} \right).$$

The case $i = t$ gives:

$$u_t^* = -(R + B^\top P B)^{-1} B^\top \left( P A x_t + \sum_{j=0}^{k-1} F^{\top j} P w_{t+j} \right),$$

which is the MPC policy at time step $t$, and is same as Equation (2). $\qquad\square$

## B    Proofs of Section 4

### B.1    Proof of Theorem 4.1

*The optimal control policy with general stochastic disturbance is given by:*

$$u_t = -(R + B^\top P B)^{-1} B^\top \left( P A x_t + \sum_{i=0}^{k-1} F^{\top i} P w_{t+i} + \sum_{i=k}^{\infty} F^{\top i} P \mu_{t+i|t+k-1} \right), \qquad (4)$$

*where $\mu_{t'|t} = \mathbb{E}[w_{t'} \mid w_0, \ldots, w_t]$. Under this policy, the marginal benefit of obtaining an extra prediction decays exponentially fast in the existing number $k$ of predictions. Formally, for $k \geq 1$,*

$$\mathsf{STO}_k - \mathsf{STO}_{k+1} = O(\|F^k\|^2) = O(\lambda^{2k}).$$

*Proof.* Similar to the proof of Theorem 3.2, we assume

$$V_t(x_t; w_{0:t+k-1}) = x_t^\top P_t x_t + x_t^\top v_t + q_t,$$

where $V_t$ has a similar definition as in Equation (5) but may further depend on $w_0, \ldots, w_{t-1}$ because the disturbance sequence is no longer Markovian. In this case, $P_t$, $v_t$ and $q_t$ still satisfy the recursive

forms in Equation (7). However, the expected values of $w_t$ and $v_t$ are different since we have a more general distribution now. Let $T - t \to \infty$, $\mu_{t'|t} = \mathbb{E}[w_{t'} \mid w_0, \ldots, w_t]$ and $F = A - HPA$. Then,

$$v_t^k = 2\sum_{i=0}^{k-1} F^{\top i+1} P w_{t+i} + 2\sum_{i=k}^{\infty} F^{\top i+1} P \mu_{t+i|t+k-1}, \tag{12}$$

$$q_t^k = w_t^\top (P - PHP) w_t + w_t^\top (I - PH) \mathop{\mathbb{E}}_{w_{t+k}} \left[ v_{t+1}^k \right] - \frac{1}{4} \mathop{\mathbb{E}}_{w_{t+k}} \left[ v_{t+1}^k \right]^\top H \mathop{\mathbb{E}}_{w_{t+k}} \left[ v_{t+1}^k \right] + \mathop{\mathbb{E}}_{w_{t+k}} \left[ q_{t+1}^k \right],$$

where the superscript $k$ denotes the number of predictions.

The optimal policy in this case has the same form as Equation (6). Plugging Equation (12) into it, we obtain the optimal policy in the theorem.

Further,

$$\mathbb{E}\left[ q_t^k - q_t^{k+1} \right] = \mathbb{E}\left[ w_t^\top (I - PH) \left( \mathop{\mathbb{E}}_{w_{t+k}} \left[ v_{t+1}^k \right] - \mathop{\mathbb{E}}_{w_{t+k+1}} \left[ v_{t+1}^{k+1} \right] \right) \right] \tag{13a}$$

$$+ \frac{1}{4} \mathbb{E}\left[ \mathop{\mathbb{E}}_{w_{t+k+1}} \left[ v_{t+1}^{k+1} \right]^\top H \mathop{\mathbb{E}}_{w_{t+k+1}} \left[ v_{t+1}^{k+1} \right] - \mathop{\mathbb{E}}_{w_{t+k}} \left[ v_{t+1}^k \right]^\top H \mathop{\mathbb{E}}_{w_{t+k}} \left[ v_{t+1}^k \right] \right] \tag{13b}$$

$$+ \mathbb{E}\left[ q_{t+1}^k - q_{t+1}^{k+1} \right], \tag{13c}$$

where the expectation $\mathbb{E}$ is taken over all randomness. Part (13a) is zero because

$$\mathop{\mathbb{E}}_{w_{t+k}} \left[ v_{t+1}^k \right] = \mathop{\mathbb{E}}_{w_{t+k},w_{t+k+1}} \left[ v_{t+1}^{k+1} \right].$$

$$\text{Part (13b)} = \frac{1}{4} \mathop{\mathbb{E}}_{w_{t+k}} \left[ \left( \mathop{\mathbb{E}}_{w_{t+k+1}} \left[ v_{t+1}^{k+1} \right] - \mathop{\mathbb{E}}_{w_{t+k}} \left[ v_{t+1}^k \right] \right)^\top H \left( \mathop{\mathbb{E}}_{w_{t+k+1}} \left[ v_{t+1}^{k+1} \right] - \mathop{\mathbb{E}}_{w_{t+k}} \left[ v_{t+1}^k \right] \right) \right]$$

$$= \mathop{\mathbb{E}}_{w_{t+k}} \left[ z_{k,t}^\top H z_{k,t} \right],$$

where

$$z_{k,t} = F^{\top k} P (w_{t+k} - \mu_{t+k|t+k-1}) + \sum_{i=k+1}^{\infty} F^{\top i} P (\mu_{t+i|t+k} - \mu_{t+i|t+k-1}).$$

Note that $z_{k,t} = F^\top z_{k-1,t+1} = F^{\top k} z_{0,t+k}$. Thus,

$$\mathsf{STO}_k - \mathsf{STO}_{k+1} = \lim_{T \to \infty} \frac{1}{T} \mathbb{E}\left[ q_0^k - q_0^{k+1} \right]$$

$$= \lim_{T \to \infty} \frac{1}{T} \sum_{t=0}^{T-1} \mathbb{E}\left[ z_{k,t}^\top H z_{k,t} \right]$$

$$= \lim_{T \to \infty} \frac{1}{T} \sum_{t=0}^{T-1} \mathbb{E}\left[ z_{0,t+k}^\top F^k H F^{\top k} z_{0,t+k} \right]$$

$$= \lim_{T \to \infty} \frac{1}{T} \sum_{t=0}^{T-1} \operatorname{Tr}\left\{ F^k H F^{\top k} \mathbb{E}\left[ z_{0,t+k} z_{0,t+k}^\top \right] \right\}$$

$$\leq \left\| F^k \right\|^2 \|H\| \lim_{T \to \infty} \frac{1}{T} \sum_{t=0}^{T-1} \operatorname{Tr} \mathbb{E}\left[ z_{0,t+k} z_{0,t+k}^\top \right]$$

where in the last line we use the fact that if $A$ is symmetric, then $\operatorname{Tr}\{AB\} \leq \lambda_{\max}(A) \operatorname{Tr}\{B\}$. Finally we just need to show the last item $\operatorname{Tr} \mathbb{E}\left[ z_{0,t+k} z_{0,t+k}^\top \right]$ is uniformly bounded for all $t$. This is straightforward because the cross-correlation of each disturbance pair is uniformly bounded, i.e., there exists $m > 0$ such that for all $t, t' \geq 1$, $\mathbb{E}\left[ w_t^\top w_{t'} \right] \leq m$.

$$\operatorname{Tr} \mathbb{E}\left[ z_{0,t} z_{0,t}^\top \right] = \sum_{i,j=0}^{\infty} \operatorname{Tr} \mathbb{E}\left[ P F^i F^{\top j} P (\mu_{t+j|t} - \mu_{t+j|t-1})(\mu_{t+i|t} - \mu_{t+i|t-1})^\top \right]$$

$$= \sum_{i,j=0}^{\infty} \mathrm{Tr}\left\{ PF^i F^{\top j} P\, \mathbb{E}\left[ \mu_{t+j|t}\mu_{t+i|t}^{\top} - \mu_{t+j|t-1}\mu_{t+i|t-1}^{\top} \right] \right\}$$

$$\leq \sum_{i,j=0}^{\infty} \left\| F^i \right\| \left\| F^j \right\| \|P\|^2\, \mathbb{E}\left[ w_{t+j}^{\top}w_{t+i} - w_{t+j}^{\top}w_{t+i} \right]$$

$$\leq \sum_{i,j=0}^{\infty} c\lambda^i c\lambda^j \|P\|^2 2m = 2\frac{c^2}{(1-\lambda)^2}\|P\|^2 m$$

for some constant $c$ from Gelfand's formula. Thus $\mathrm{Tr}\,\mathbb{E}\left[ z_{0,t}z_{0,t}^{\top} \right]$ is bounded by a constant independent of $t$. Thus,

$$\mathsf{STO}_k - \mathsf{STO}_{k+1} = O(\|F^k\|^2).$$

$\square$

## B.2  Proof of Theorem 4.4

$\mathsf{MPCS}_k - \mathsf{MPCS}_{k+1} = O(\|F^k\|^2) = O(\lambda^{2k})$. *Moreover, in Example 4.3,* $\mathsf{MPCS}_k - \mathsf{MPCS}_{k+1} = \Theta(\|F^k\|^2)$.

*Proof.* To recursively calculate the value of $J^{\mathsf{MPC}_k}$, we define:

$$V_t^{\mathsf{MPC}_k}(x_t; w_{0:t+k-1}) = \sum_{i=t}^{T-1}(x_i^{\top}Qx_i + u_i^{\top}Ru_i) + x_T^{\top}Q_f x_T$$
$$= x_t^{\top}Qx_t + u_t^{\top}Ru_t + V_{t+1}(Ax_t + Bu_t + w_t; w_{0:t+k})$$

as the cost-to-go function with MPC as the policy, i.e., $u_t$ is the control at time step $t$ from the MPC policy with $k$ predictions. Similar to the previous proofs, we assume $V_t^{\mathsf{MPC}_k}(x) = x^{\top}P_t x + x^{\top}v_t + q_t$ (which turns out to be correct by induction) and $T - t \to \infty$ so that $P_t = P$. Then,

$$V_t^{\mathsf{MPC}_k}(x_t; w_{0:t+k-1}) = x_t^{\top}Qx_t + u_t^{\top}Ru_t + (Ax_t + Bu_t + w_t)^{\top}P(Ax_t + Bu_t + w_t)$$
$$+ (Ax_t + Bu_t + w_t)^{\top}v_{t+1} + q_{t+1}$$
$$= u_t^{\top}(R + B^{\top}PB)u_t + 2u_t^{\top}B^{\top}(PAx_t + Pw_t + v_{t+1}/2)$$
$$+ x_t^{\top}Qx_t + (Ax_t + w_t)^{\top}P(Ax_t + w_t) + (Ax_t + w_t)^{\top}v_{t+1} + q_{t+1}. \tag{14}$$

Let $F = A - HPA$. Plugging in the formula of $u_t$ in Theorem 3.3, we have

$$V_t^{\mathsf{MPC}_k}(x_t; w_{0:t+k-1}) = \left( \frac{1}{2}v_{t+1} - \sum_{i=1}^{k-1}F^{\top i}Pw_{t+i} \right)^{\top} H \left( \frac{1}{2}v_{t+1} - \sum_{i=1}^{k-1}F^{\top i}Pw_{t+i} \right)$$
$$- \left( PAx_t + Pw_t + \frac{1}{2}v_{t+1} \right)^{\top} H \left( PAx_t + Pw_t + \frac{1}{2}v_{t+1} \right)$$
$$+ x_t^{\top}Qx_t + (Ax_t + w_t)^{\top}P(Ax_t + w_t) + (Ax_t + w_t)^{\top}v_{t+1} + q_{t+1}$$
$$= x_t^{\top}(Q + A^{\top}PA - A^{\top}PHPA)x_t + x_t^{\top}(F^{\top}v_{t+1} + 2F^{\top}Pw_t)$$
$$+ \left( \frac{1}{2}v_{t+1} - \sum_{i=1}^{k-1}F^{\top i}Pw_{t+i} \right)^{\top} H \left( \frac{1}{2}v_{t+1} - \sum_{i=1}^{k-1}F^{\top i}Pw_{t+i} \right)$$
$$- \left( Pw_t + \frac{1}{2}v_{t+1} \right)^{\top} H \left( Pw_t + \frac{1}{2}v_{t+1} \right) + w_t^{\top}Pw_t + w_t^{\top}v_{t+1} + q_{t+1}$$
$$= x_t^{\top}Px_t + x_t^{\top}v_t + q_t.$$

Thus,

$$v_t = F^{\top}v_{t+1} + 2F^{\top}Pw_t = 2\sum_{i=0}^{\infty}F^{\top i+1}Pw_{t+i}.$$

Then, we can plug $v_{t+1}$ into $q_t$:

$$q_t = q_{t+1} + \left(\sum_{i=k}^{\infty} F^{\top i} P w_{t+i}\right)^{\top} H \left(\sum_{i=k}^{\infty} F^{\top i} P w_{t+i}\right)$$

$$- \left(\sum_{i=0}^{\infty} F^{\top i} P w_{t+i}\right)^{\top} H \left(\sum_{i=0}^{\infty} F^{\top i} P w_{t+i}\right) + w_t^{\top} P w_t + 2 w_t^{\top} \left(\sum_{i=1}^{\infty} F^{\top i} P w_{t+i}\right). \quad (15)$$

Note that Equation (15) is for MPC with $k$ predictions. With the disturbance sequence $\{w_t\}$ fixed, we can compare the per-step cost of MPC with $k$ predictions and that with $k+1$ predictions:

$$q_t^k - q_t^{k+1} = q_{t+1}^k - q_{t+1}^{k+1} + \left(\sum_{i=k}^{\infty} F^{\top i} P w_{t+i}\right)^{\top} H \left(\sum_{i=k}^{\infty} F^{\top i} P w_{t+i}\right)$$

$$- \left(\sum_{i=k+1}^{\infty} F^{\top i} P w_{t+i}\right)^{\top} H \left(\sum_{i=k+1}^{\infty} F^{\top i} P w_{t+i}\right)$$

$$= q_{t+1}^k - q_{t+1}^{k+1} + w_{t+k}^{\top} P F^k H F^{\top k} \left(P w_{t+k} + 2 \sum_{i=1}^{\infty} F^{\top i} P w_{t+i+k}\right). \quad (16)$$

Thus,

$$\mathbb{E}\left[q_t^k - q_t^{k+1} - (q_{t+1}^k - q_{t+1}^{k+1})\right] = \mathbb{E}\left[w_{t+k}^{\top} P F^k H F^{\top k} \left(P w_{t+k} + 2 \sum_{i=1}^{\infty} F^{\top i} P w_{t+i+k}\right)\right]$$

$$= \mathrm{Tr}\left\{P F^k H F^{\top k} \left(P \mathbb{E}\left[w_{t+k} w_{t+k}^{\top}\right] + 2 \sum_{i=1}^{\infty} F^{\top i} P \mathbb{E}\left[w_{t+i+k} w_{t+k}^{\top}\right]\right)\right\}$$

$$= \mathrm{Tr}\left\{P F^k H F^{\top k} Z_{k,t}\right\},$$

where $Z_{k,t} = P \mathbb{E}\left[w_{t+k} w_{t+k}^{\top}\right] + 2 \sum_{i=1}^{\infty} F^{\top i} P \mathbb{E}\left[w_{t+i+k} w_{t+k}^{\top}\right]$. Note that $Z_{k,t} = Z_{k-1,t+1}$.

$$\mathsf{MPCS}_k - \mathsf{MPCS}_{k+1} = \lim_{T \to \infty} \frac{1}{T} \mathbb{E}\left[q_0^k - q_0^{k+1}\right]$$

$$= \lim_{T \to \infty} \frac{1}{T} \sum_{t=0}^{T-1} \mathrm{Tr}\left\{P F^k H F^{\top k} Z_{k,t}\right\}$$

$$\leq \lim_{T \to \infty} \frac{1}{T} \sum_{t=0}^{T-1} \|P\| \|H\| \|F^k\|^2 \mathrm{Tr}\{Z_{k,t}\},$$

where in the last line we use the fact that if $A$ is symmetric, then $\mathrm{Tr}\{AB\} \leq \|A\| \mathrm{Tr}\{B\}$. Similarly to the last part in the proof of Theorem 4.1, now we just need to show the last term $\mathrm{Tr}\{Z_{k,t}\}$ is uniformly bounded for all $t$. Again, this is because the cross-correlation of each disturbance pair is uniformly bounded.

$$\mathrm{Tr}\{Z_{k,t}\} \leq \|P\| \mathrm{Tr}\, \mathbb{E}\left[w_{t+k} w_{t+k}^{\top}\right] + 2 \sum_{i=1}^{\infty} \|P\| \|F^i\| \, \mathbb{E}\left[\sum_j \sigma_j(w_{t+i+k} w_{t+k}^{\top})\right]$$

$$\leq \|P\| m + 2 \sum_{i=1}^{\infty} c \lambda^i \|P\| m = \|P\| m + 2c \frac{\lambda}{1-\lambda} \|P\| m$$

where $c$ is some constant, and in the first line, we use the fact that $\mathrm{Tr}\{AB\} \leq \|A\| \sum_j \sigma_j(B)$ with $\sigma_j(\cdot)$ denoting the $j$-th singular value. Thus, $\mathrm{Tr}\{Z_{k,t}\}$ is uniformly bounded. Therefore, $\mathsf{MPCS}_k - \mathsf{MPCS}_{k+1} = O(\|F^k\|^2)$. $\qquad \square$

## B.3 Proof of Theorem 4.6

$Reg^S(\mathsf{MPC}_k) = MPCS_k^T - STO_T^T = O(\|F^k\|^2 T + 1) = O(\lambda^{2k}T + 1)$, *where the second term results from the difference between finite/infinite horizons.*

*Proof.* To calculate the dynamic regret, we cannot simply let $T - t \to \infty$ as we did before Equation (14) in the proof of Theorem 4.4 and instead need to handle the expressions in a more delicate manner. In particular, we need to rigorously analyze the impact of finite horizon. Let $\Delta_t = P_t - P$.

$$
\begin{aligned}
&V_t^{\mathsf{MPC}_k}(x_t; w_{0:t+k-1}) \\
&= u_t^\top(R + B^\top P_{t+1}B)u_t + 2u_t^\top B^\top(P_{t+1}Ax_t + P_{t+1}w_t + v_{t+1}/2) \\
&\quad + x_t^\top Q x_t + (Ax_t + w_t)^\top P_{t+1}(Ax_t + w_t) + (Ax_t + w_t)^\top v_{t+1} + q_{t+1} \\
&= u_t^\top(R + B^\top PB)u_t + 2u_t^\top B^\top(PAx_t + Pw_t + v_{t+1}/2) \\
&\quad + x_t^\top Q x_t + (Ax_t + w_t)^\top P(Ax_t + w_t) + (Ax_t + w_t)^\top v_{t+1} + q_{t+1} \\
&\quad + u_t^\top B^\top \Delta_{t+1}Bu_t + 2u_t^\top B^\top \Delta_{t+1}(Ax_t + w_t) + (Ax_t + w_t)^\top \Delta_{t+1}(Ax_t + w_t).
\end{aligned}
$$

Plugging in the MPC policy as in Theorem 3.3, we have:

$$
\begin{aligned}
&V_t^{\mathsf{MPC}_k}(x_t; w_{0:t+k-1}) \\
&= x_t^\top(Q + A^\top PA - A^\top PHPA)x_t + x_t^\top(F^\top v_{t+1} + 2F^\top Pw_t) \\
&\quad + \left(\frac{1}{2}v_{t+1} - \sum_{i=1}^{k-1}F^{\top i}Pw_{t+i}\right)^\top H\left(\frac{1}{2}v_{t+1} - \sum_{i=1}^{k-1}F^{\top i}Pw_{t+i}\right) \\
&\quad - \left(Pw_t + \frac{1}{2}v_{t+1}\right)^\top H\left(Pw_t + \frac{1}{2}v_{t+1}\right) + w_t^\top Pw_t + w_t^\top v_{t+1} + q_{t+1} \\
&\quad + \left(Fx_t + w_t - \sum_{i=0}^{k-1}F^{\top i}Pw_{t+i}\right)^\top \Delta_{t+1}\left(Fx_t + w_t - \sum_{i=0}^{k-1}F^{\top i}Pw_{t+i}\right) \\
&= x_t^\top(Q + A^\top PA - A^\top PHPA + F^\top \Delta_{t+1}F)x_t \\
&\quad + x_t^\top\left(F^\top v_{t+1} + 2F^\top Pw_t + 2F^\top \Delta_{t+1}\left(w_t - \sum_{i=0}^{k-1}F^{\top i}Pw_{t+i}\right)\right) \\
&\quad + \left(\frac{1}{2}v_{t+1} - \sum_{i=1}^{k-1}F^{\top i}Pw_{t+i}\right)^\top H\left(\frac{1}{2}v_{t+1} - \sum_{i=1}^{k-1}F^{\top i}Pw_{t+i}\right) \\
&\quad - \left(Pw_t + \frac{1}{2}v_{t+1}\right)^\top H\left(Pw_t + \frac{1}{2}v_{t+1}\right) + w_t^\top Pw_t + w_t^\top v_{t+1} + q_{t+1} \\
&\quad + \left(w_t - \sum_{i=0}^{k-1}F^{\top i}Pw_{t+i}\right)^\top \Delta_{t+1}\left(w_t - \sum_{i=0}^{k-1}F^{\top i}Pw_{t+i}\right)
\end{aligned}
$$

Comparing this with the induction hypothesis $V_t^{\mathsf{MPC}_k} = x_t^\top(P + \Delta_t)x_t + x_t^\top v_t + q_t$, we obtain the recursive formulae for $\Delta_t, v_t, q_t$.

$$
\Delta_t = F^\top \Delta_{t+1}F = F^{\top T-t}\Delta_T F^{T-t} = F^{\top T-t}(Q_f - P)F^{T-t}.
$$

This implies that $P_t$ converges to $P$ exponentially fast, i.e., $\|\Delta_t\| = O(\|F^{T-t}\|^2) = O(\lambda^{2(T-t)})$.

$$
\begin{aligned}
v_t &= F^\top v_{t+1} + 2F^\top Pw_t + 2F^\top \Delta_{t+1}\left(w_t - \sum_{i=0}^{k-1}F^{\top i}Pw_{t+i}\right) \\
&= 2\sum_{j=0}^{T-t-1}\left(F^{\top j+1}Pw_{t+j} + F^{\top j+1}\Delta_{t+j+1}\left(w_{t+j} - \sum_{i=0}^{k-1}F^{\top i}Pw_{t+j+i}\right)\right)
\end{aligned}
$$

$$= 2\sum_{i=0}^{T-t-1} F^{\top i+1} P w_{t+i} + 2\sum_{j=0}^{T-t-1} F^{\top j+1}\Delta_{t+j+1}\left(w_{t+j} - \sum_{i=0}^{k-1} F^{\top i} P w_{t+j+i}\right).$$

Denote the second term by $2d_t$. We have

$$d_t = \sum_{j=0}^{T-t-1} F^{\top j+1}\Delta_{t+j+1}\left(w_{t+j} - \sum_{i=0}^{k-1} F^{\top i} P w_{t+j+i}\right)$$

$$= \sum_{j=0}^{T-t-1} O(\lambda^j \lambda^{2(T-t-j)}) = O(\lambda^{T-t}).$$

$$d_t^k - d_t^{k+1} = \sum_{j=0}^{T-t-k-1} F^{\top j+1}\Delta_{t+j+1} F^{\top k} P w_{t+j+k} \tag{17}$$

$$= \sum_{j=0}^{T-t-k-1} O(\lambda^j \lambda^{2(T-t-j)}\|F^k\|) = O(\lambda^{T-t+k}\|F^k\|).$$

Finally, we have a formula for $q_t$ that parallels Equation (15):

$$q_t = q_{t+1} + \left(d_{t+1} + \sum_{i=k}^{T-t-1} F^{\top i} P w_{t+i}\right)^{\top} H\left(d_{t+1} + \sum_{i=k}^{T-t-1} F^{\top i} P w_{t+i}\right)$$

$$- \left(d_{t+1} + \sum_{i=0}^{T-t-1} F^{\top i} P w_{t+i}\right)^{\top} H\left(d_{t+1} + \sum_{i=0}^{T-t-1} F^{\top i} P w_{t+i}\right)$$

$$+ w_t^{\top} P w_t + 2 w_t^{\top}\left(d_{t+1} + \sum_{i=1}^{T-t-1} F^{\top i} P w_{t+i}\right).$$

Taking the difference between $k$ and $k+1$ predictions, we have

$$q_t^k - q_t^{k+1} - (q_{t+1}^k - q_{t+1}^{k+1})$$

$$= (w_{t+k}^{\top} P F^k + (d_{t+1}^k - d_{t+1}^{k+1})^{\top}) H\left(d_{t+1}^k + d_{t+1}^{k+1} + F^{\top k} P w_{t+k} + 2\sum_{i=1}^{T-t-k-1} F^{\top i+k} P w_{t+i+k}\right)$$
$$\tag{18}$$

$$= (w_{t+k}^{\top} P F^k + O(\lambda^{T-t}\|F^k\|)) H\left(O(\lambda^{T-t}) + F^{\top k} P w_{t+k} + 2\sum_{i=1}^{T-t-k-1} F^{\top i+k} P w_{t+i+k}\right),$$

and thus

$$\mathbb{E}\big[q_t^k - q_t^{k+1} - (q_{t+1}^k - q_{t+1}^{k+1})\big] = O(\|F^k\|(\lambda^{T-t} + \|F^k\|)).$$

$$\mathbb{E}\big[q_0^k - q_0^T\big] = \sum_{t=0}^{T-1} \mathbb{E}\big[q_t^k - q_t^{k+1} - (q_{t+1}^k - q_{t+1}^{k+1})\big]$$

$$= \sum_{t=0}^{T-1} O(\|F^k\|(\lambda^{T-t} + \|F^k\|))$$

$$= O(\|F^k\|^2 T + \|F^k\|).$$

$$\mathbb{E}\big[v_0^k - v_0^T\big] = 2(d_0^k - d_0^T) = O(\lambda^{T+k}\|F^k\|).$$

$$\mathbb{E}\, J^{\mathsf{MPC}_k} - \mathbb{E}\, J^{\mathsf{MPC}_T} = \mathbb{E}\big[V_0^k(x_0) - V_0^T(x_0)\big]$$
$$= \mathbb{E}\big[x_0^{\top}(v_0^k - v_0^T) + (q_0^k + q_0^T)\big]$$

$$= O(\|F^k\|^2 T + \|F^k\|). \tag{19}$$

By definition, $J^{\mathsf{MPC}_T}$ is the cost of MPC policy given all future disturbances before making any decisions. It almost equals to $\min_u J$, the optimal policy given all future disturbances, except that during optimization, MPC assumes the final-step cost to be $x_T^\top P x_T$ instead of $x_T^\top Q_f x_T$. This will incur at most constant extra cost, i.e.,

$$J^{\mathsf{MPC}_T} - \min_u J = O(P - Q_f) = O(1). \tag{20}$$

By Equations (19) and (20),

$$Reg^S(\mathsf{MPC}_k) = \mathbb{E}\, J^{\mathsf{MPC}_k} - \mathbb{E} \min_u J = O(\|F^k\|^2 T + \|F^k\| + 1) = O(\|F^k\|^2 T + 1).$$

$\square$

### B.4   Proof of Theorem 4.7

*The optimal dynamic regret $Reg_k^{S^*} = STO_k^T - STO_T^T = O(\|F^k\|^2 T + 1) = O(\lambda^{2k} T + 1)$ and there exist $A$, $B$, $Q$, $R$, $Q_f$, $x_0$, and $\mathcal{W}$ such that $Reg_k^{S^*} = \Theta(\|F^k\|^2(T - k))$.*

*Proof.* The first part follows from Theorem 4.6 and that fact that $Reg_k^{S^*} \le Reg^S(\mathsf{MPC}_k)$.

The second part is shown by Example 4.3, i.e., suppose $n = d = 1$ and the disturbance are i.i.d. and zero-mean. Additionally, let $Q_f = P$ and $x_0 = 0$. In this case, MPC has not only the same policy but also the same cost as the optimal control policy. Also, $P_t = P$ for all $t$. To calculate the total cost, we follow the approach used in the proof of Theorem 3.2. Since $T$ is finite now, we have a similar (to Equation (8)) but different form of $v_t$:

$$v_t = 2 \sum_{i=0}^{\min\{k-1, T-t-1\}} F^{\top\,i+1} P w_{t+i}.$$

Thus,

$$\mathbb{E}[q_t] = \mathrm{Tr}\left\{ \left( P - \sum_{i=0}^{\min\{k-1, T-t-1\}} P F^i H F^{\top\,i} P \right) W \right\} + \mathbb{E}[q_{t+1}].$$

$$\mathbb{E}[q_0] = \mathrm{Tr}\left\{ \sum_{t=0}^{T-1} \left( P - \sum_{i=0}^{\min\{k-1, T-t-1\}} P F^i H F^{\top\,i} P \right) W \right\}.$$

Let $q_t^k$ denote $q_t$ in the scenario of $k$ predictions.

$$Reg^{S^*} = \mathbb{E}[q_0^k - q_0^T] = \mathrm{Tr}\left\{ \sum_{t=0}^{T-k-1} \sum_{i=k}^{T-t-1} P F^i H F^{\top\,i} P W \right\}$$

$$\ge (T - k)\,\mathrm{Tr}\left\{ P F^k H F^{\top\,k} P W \right\} = \Omega(\|F^k\|^2(T - k)).$$

On the other hand,

$$Reg^{S^*} = \mathbb{E}[q_0^k - q_0^T] \le (T - k)\,\mathrm{Tr}\left\{ \sum_{i=k}^{\infty} P F^i H F^{\top\,i} P W \right\} = O(\|F^k\|^2(T - k)).$$

Therefore, $Reg^{S^*} = \Theta(\|F^k\|^2(T - k))$.　　$\square$

## C   Proofs of Section 5

### C.1   Proof of Theorem 5.1

*For $k \ge 1$, $\mathsf{ADV}_k - \mathsf{ADV}_{k+1} = O(\|F^k\|^2) = O(\lambda^{2k})$.*

*Proof.* This proof is based on Theorem 5.3. It turns out that the behavior of the MPC policy and its cost is easier to analyze than the optimal one, especially in the adversarial setting.

$$\mathsf{ADV}_k - \mathsf{ADV}_{k+1} \leq \mathsf{ADV}_k - \mathsf{ADV}_\infty \leq \mathsf{MPCA}_k - \mathsf{ADV}_\infty = \sum_{i=k}^{\infty} \mathsf{MPCA}_i - \mathsf{MPCA}_{i+1}.$$

By Theorem 5.3,

$$\mathsf{MPCA}_i - \mathsf{MPCA}_{i+1} \leq O\left(\left\|F^i\right\|^2\right) \leq O\left(\left\|F^k\right\|^2 \left\|F^{i-k}\right\|^2\right) \leq O\left(\left\|F^k\right\|^2 \lambda^{2(i-k)}\right).$$

Thus,

$$\mathsf{ADV}_k - \mathsf{ADV}_{k+1} \leq O\left(\left\|F^k\right\|^2 \sum_{i=k}^{\infty} \lambda^{2(i-k)}\right) = O(\|F^k\|^2).$$

□

### C.2 Proof of Example 5.2

*Let $A = B = Q = R = 1$ and $\Omega = [-1, 1]$. In this case, one prediction is enough to leverage the full power of prediction. Formally, we have $\mathsf{ADV}_1 = \mathsf{ADV}_\infty = 1$. In other words, for all $k \geq 1$, $\mathsf{ADV}_k = 1$. The optimal control policy (as $T \to \infty$) is a piecewise function:*

$$u^*(x, w) = \begin{cases} -(x+w) & , -1 \leq x+w \leq 1 \\ -(x+w) + \frac{3-\sqrt{5}}{2}(x+w-1) & , x+w > 1 \\ -(x+w) + \frac{3-\sqrt{5}}{2}(x+w+1) & , x+w < -1 \end{cases}.$$

*The proof leverages two different cost-to-go functions for the* min *player and the* sup *player.*

*Proof.* We will show $\mathsf{ADV}_1 = 1$ and $\mathsf{ADV}_\infty = 1$ separately. The system dynamics is given by $x_{t+1} = x_t + u_t + w_t$ with $w_t \in [-1, 1]$ and

$$ADV_1^T = \max_{w_0} \min_{u_0} \cdots \max_{w_{T-1}} \min_{u_{T-1}} \sum_{t=0}^{T-1} (x_t^2 + u_t^2) + x_T^2.$$

We will calculate the results of each min and max by dynamical programming. In particular, we will define two cost-to-go functions for the min player and the max player respectively. Let $z_t = x_t + w_t$. Then, $z_t$ can be regarded as the disturbed state. This is natural since the controller has one prediction and decides $u_t$ after knowing $w_t$. Thus, the system dynamics can be split into two stages: $z_t = x_t + w_t$ and $x_{t+1} = z_t + u_t$. Let

$$f_t(z_t) = \min_{u_t} \max_{w_{t+1}} \min_{u_{t+1}} \cdots \max_{w_{T-1}} \min_{u_{T-1}} \sum_{i=t}^{T-1} (u_i^2 + x_{i+1}^2)$$

$$= \min_{u_t} \left(u_t^2 + (z_t + u_t)^2 + g_{t+1}(z_t + u_t)\right),$$

$$g_t(x_t) = \max_{w_t} \min_{u_t} \cdots \max_{w_{T-1}} \min_{u_{T-1}} \sum_{i=t}^{T-1} (u_i^2 + x_{i+1}^2)$$

$$= \max_{w_t} f_t(x_t + w_t).$$

For $t = T - 1$, we have

$$f_{T-1}(z) = \min_u u^2 + (z+u)^2 = \frac{z^2}{2},$$

$$g_{T-1}(x) = \max_w \frac{(x+w)^2}{2} = \frac{(|x|+1)^2}{2}.$$

We will prove by backward induction that $g_t(x) = a_t x^2 + 2b_t |x| + c_t$ where $a_t, b_t, c_t$ are some coefficients with $0 < b_t < 1$. Assuming this is true at $t$, we will show this is true at $t - 1$.

$$f_{t-1}(z) = \min_u \left(u^2 + (z+u)^2 + g_t(z+u)\right)$$

$$= \min_y \big((y-z)^2 + y^2 + g_t(y)\big)$$

$$= \min_y \big((y-z)^2 + y^2 + a_t y^2 + 2b_t|y| + c_t\big)$$

$$= \min_y \big((a_t + 2)y^2 - 2(z - b_t \operatorname{sign}(y))y + z^2 + c_t\big),$$

where $y = z + u = x + w + u$ is the state after the control policy is applied. Let function $y(z)$ map from the disturbed old state to the new state. The optimal $y$ is given by:

$$y^*(z) = \arg\min_y \big((a_t + 2)y^2 - 2(z - b_t \operatorname{sign}(y))y + z^2 + c_t\big)$$

$$= \begin{cases} 0 & , -b_t \le z \le b_t \\ \frac{z - b_t \operatorname{sign}(z)}{a_t + 2} & , \text{ otherwise} \end{cases}. \tag{21}$$

Thus, for $z < -b_t$ or $z > b_t$, we have

$$f_{t-1}(z) = -\frac{(z - b_t \operatorname{sign}(z))^2}{a_t + 2} + z^2 + c_t$$

$$= -\frac{z^2 - 2b_t|z| + b_t^2}{a_t + 2} + z^2 + c_t$$

$$= \frac{a_t + 1}{a_t + 2}z^2 + \frac{2b_t}{a_t + 2}|z| + c_t - \frac{b_t^2}{a_t + 2}.$$

For $z \in [-b_t, b_t]$, the value of $f_t(z)$ is not needed in the calculation of $g_t(x)$ because $0 < b_t < 1$ (induction hypothesis) and the adversary — who wants to maximize $f_t(z_t)$, a convex, even function — will never choose $w_t$ such that $z_t = x_t + w_t \in (-1, 1)$ since $w_t$ can be chosen from $[-1, 1]$.

$$g_{t-1}(x) = \max_w f_t(x + w) = f_t(x + \operatorname{sign}(x))$$

$$= \frac{a_t + 1}{a_t + 2}(x^2 + 2|x| + 1) + \frac{2b_t}{a_t + 2}(|x| + 1) + c_t - \frac{b_t^2}{a_t + 2}$$

$$= \frac{a_t + 1}{a_t + 2}x^2 + \frac{2(a_t + b_t + 1)}{a_t + 2}|x| + c_t + \frac{a_t + 1 + 2b_t - b_t^2}{a_t + 2}$$

$$= a_{t-1}x^2 + 2b_{t-1}|x| + c_{t-1}.$$

Now, we have obtained the recursive formulae for $a_t, b_t, c_t$. The initial values are $a_{T-1} = b_{T-1} = c_{T-1} = 1/2$.

Let $\mathfrak{f}_i$ be the $i$-th Fibonacci number with $\mathfrak{f}_0 = 0, \mathfrak{f}_1 = 1$. Then, $a_{T-i} = \mathfrak{f}_{i+1}/\mathfrak{f}_{i+2}$. As $i \to \infty$, $a_{T-i} \to \frac{\sqrt{5}-1}{2}$.

For $b_t$, we have $1 - b_{T-(i+1)} = (1 - b_{T-i})/(a_{T-i} + 2)$. When $i$ is large, $1 - b_{T-i}$ approaches $0$ but is always positive. Thus, $b_{T-i}$ approaches $1$ but is always less than $1$.

For $c_t$, we have

$$c_{T-(i+1)} = c_{T-i} + 1 - \frac{(1 - b_{T-i})^2}{a_{T-i} + 2}$$

and thus $c_{T-(i+1)} - c_{T-i} \to 1$. Therefore, $\mathsf{ADV}_1 = 1$.

The optimal control policy is obtained by plugging the above values back into Equation (21):

$$u^*(x, w) = -(x + w) + y^*(x + w) = -(x + w) + \begin{cases} 0 & , -1 \le x + w \le 1 \\ \frac{x + w - \operatorname{sign}(x+w)}{\frac{\sqrt{5}+3}{2}} & , \text{ otherwise} \end{cases}.$$

For $\mathsf{ADV}_\infty$, we will show that $\mathsf{STO}_\infty = 1$ at a specific disturbance sequence: $w_t = 1$ for all $t$. Because $\mathsf{STO}_\infty \le \mathsf{ADV}_\infty \le \mathsf{ADV}_1 = 1$, we know that $\mathsf{ADV}_\infty = 1$.

According to Equations (8) and (9) with $k \to \infty$,

$$\mathsf{STO}_\infty = \lim_{T \to \infty} \frac{1}{T}\sum_{t=0}^{T-1}(2w_t\psi_t - Pw_t^2 - H\psi_t^2) \text{ with } \psi_t = \sum_{i=0}^{\infty} F^i P w_{t+i}.$$

Solving the Riccati equation, we have $P = \frac{1+\sqrt{5}}{2}$, $H = F = \frac{3-\sqrt{5}}{2}$. When $w_t = 1$ for all $t$, $\mathsf{STO}_\infty = 1$. $\qquad\square$

## C.3 Proof of Theorem 5.3

$\mathsf{MPCA}_k - \mathsf{MPCA}_{k+1} = O(\|F^k\|^2) = O(\lambda^{2k})$.

*Proof.* Note that Equation (16) in the proof of Theorem 4.4 does not rely on the type of disturbance, i.e., Equation (16) holds for adversarial disturbance as well. Let $r = \sup_{w \in \Omega} \|w\|_2$.

$$
\begin{aligned}
q_t^k - q_t^{k+1} - (q_{t+1}^k - q_{t+1}^{k+1}) &= w_{t+k}^\top P F^k H F^{\top k} \left( P w_{t+k} + 2 \sum_{i=1}^\infty F^{\top i} P w_{t+i+k} \right) \\
&\leq \|w_{t+k}\| \|P\| \|H\| \|F^k\|^2 \left( \|P\| \|w_{t+k}\| + 2 \sum_{i=1}^\infty \|F^i\| \|P\| \|w_{t+i+k}\| \right) \\
&\leq \|F^k\|^2 \left( 1 + 2 \sum_{i=1}^\infty \|F^i\| \right) \|H\| \|P\|^2 r^2 \\
&\leq \|F^k\|^2 \left( 1 + 2 \frac{c\lambda}{1 - \lambda} \right) \|H\| \|P\|^2 r^2
\end{aligned}
$$

for some constant $c$.

$$
\begin{aligned}
\mathsf{MPCA}_k - \mathsf{MPCA}_{k+1} &= \lim_{T \to \infty} \frac{1}{T} (\max_w q_0^k - \max_w q_0^{k+1}) \\
&\leq \lim_{T \to \infty} \frac{1}{T} \max_w (q_0^k - q_0^{k+1}) \\
&\leq \lim_{T \to \infty} \frac{1}{T} \sum_{t=0}^{T-1} \max_w (q_t^k - q_t^{k+1} - (q_{t+1}^k - q_{t+1}^{k+1})) \\
&\leq \|F^k\|^2 \left( 1 + 2 \frac{c\lambda}{1 - \lambda} \right) \|H\| \|P\|^2 r^2 = O(\|F^k\|^2).
\end{aligned}
$$

$\square$

## C.4 Proof of Theorem 5.5

$Reg^A(\mathsf{MPC}_k) = O(\|F^k\|^2 T + 1) = O(\lambda^{2k} T + 1)$.

*Proof.* We follow the notations in the proof of Theorem 4.6. Equation (18) does not rely on the type of disturbance, so it holds for adversarial disturbance as well. By Equation (18) and the fact that $w_t$ is bounded, we have

$$
q_t^k - q_t^{k+1} - (q_{t+1}^k - q_{t+1}^{k+1}) = O(\|F^k\|(\lambda^{T-t} + \|F^k\|)),
$$

where the constant in the Big-Oh notation does not depend on the disturbance sequence $w$. Thus,

$$
\max_w (q_0^k - q_0^T) \leq \sum_{t=0}^{T-1} \max_w (q_t^k - q_t^{k+1} - (q_{t+1}^k - q_{t+1}^{k+1})) = O(\|F^k\|^2 T + \|F^k\|).
$$

By Equation (17) and the boundedness of $w_t$,

$$
\max_w (v_0^k - v_0^T) = 2 \max_w (d_0^k - d_0^T) = O(\lambda^{T+k} \|F^k\|).
$$

$$
\begin{aligned}
\max_w (J^{\mathsf{MPC}_k} - J^{\mathsf{MPC}_T}) = \max_w (V_0^k(x_0) - V_0^T(x_0)) &\leq \max_w (x_0^\top (v_0^k - v_0^T)) + \max_w (q_0^k - q_0^T) \\
&= O(\|F^k\|^2 T + \|F^k\|).
\end{aligned}
$$

As Equation (20), $J^{\mathsf{MPC}_T} - \min_u J = O(1)$. Thus,

$$
\begin{aligned}
Reg^A(\mathsf{MPC}_k) = \max_w (J^{\mathsf{MPC}_k} - \min_u J) &\leq \max_w (J^{\mathsf{MPC}_k} - J^{MPC_T}) + \max_w (J^{\mathsf{MPC}_T} - \min_u J) \\
&= O(\|F^k\|^2 T + \|F^k\| + 1) = O(\|F^k\|^2 T + 1).
\end{aligned}
$$

$\square$

## C.5 Proof of Theorem 5.6

$Reg_k^{A^*} = O(\|F^k\|^2 T + 1) = O(\lambda^{2k}T + 1)$. *Moreover, there exist A, B, Q, R, $Q_f$, $x_0$, and $\Omega$ such that* $Reg_k^{A^*} = \Omega(\|F^k\|^2(T-k))$.

*Proof.* The first part of the theorem follows from Theorem 5.5 and the fact that $Reg_k^{A^*} \leq Reg^A(\mathsf{MPC}_k)$.

We reduce the second part of this theorem to the second part of Theorem 4.7. Since the proof of Theorem 4.7 works for any fixed distribution of $w_t$ (with finite second moment), we can restrict that distribution to have bounded support. Denote this bounded support by $\Omega$. Then, we have

$$
\begin{aligned}
Reg_k^{A^*} &= \sup_{w_0,\cdots,w_{k-1}} \min_{u_0} \sup_{w_k} \cdots \min_{u_{T-k-1}} \sup_{w_{T-1}} \min_{u_{T-k},\cdots,u_{T-1}} \left( J(u,w) - \min_{u_0',\ldots,u_{T-1}'} J(u',w) \right) \\
&\geq \mathop{\mathbb{E}}_{w_0,\cdots,w_{k-1}} \min_{u_0} \mathbb{E}_{w_k} \cdots \min_{u_{T-k-1}} \mathbb{E}_{w_{T-1}} \min_{u_{T-k},\cdots,u_{T-1}} \left( J(u,w) - \min_{u_0',\ldots,u_{T-1}'} J(u',w) \right) \\
&= Reg_k^{S^*} = \Theta(\|F^k\|^2(T-k)).
\end{aligned}
$$

$\square$