[Reviews · NeurIPS 2020]

Review 1

Summary and Contributions: Update: Thanks for the thoughtful response. The suggested longer and more systematic review of related approaches in MPC will definitely improve this work. --- This paper studies the LQR problem with known dynamics and predictions of disturbances over a fixed lookahead, for both stochastic and adversarial disturbances. For both cases, the authors derive the form of the optimal policy using the predictions, and show that the regret decays exponentially in the prediction horizon, and furthermore that a simple MPC-like strategy is nearly optimal.

Strengths: This paper makes very nice connections between the online learning literature and model predictive control. The main ideas are clearly explained, and the claims are backed up by theory.

Weaknesses: The clarity around the MPC problem setting could be improved a bit, as could the explanation of its relation to related work. See the sections below.

Correctness: The results seem to be correct.

Clarity: Overall, this paper is well written. It would be good to better clarify the considered MPC setting as distinct from general MPC frameworks, which might consider planning horizons independent from the prediction horizon and which have a stronger focus on safety and feasibility.

Relation to Prior Work: There is a solid effort at placing these results in the context of related work, but it could be improved a bit. The MPC literature does not generally treat disturbances in the same manner as this work, instead placing a stronger focus on robustness (see e.g. https://arxiv.org/abs/2007.00930 and references within) and constraint satisfaction. The incorporation of learning to MPC can be motivated by the difficulty of characterizing safe terminal sets/costs as in [25,26], but is also often motivated by unknown dynamics, e.g. https://arxiv.org/abs/1107.2487. A clearer treatment of this related literature would improve the paper, especially articulating similarities and differences between learning costs, disturbances, and dynamics (e.g. reference tracking can be modeled with either costs or disturbances).

Reproducibility: Yes

Additional Feedback:


Review 2

Summary and Contributions: This work consider the task of controlling a linear quadratic system subject to perturbations (stochastic or otherwise), when at each time step the next few perturbations are made available. The performance measures considered here is regret wrt the best posthoc control sequence (best U given all perturbations; capturing the statistical challenge). The proposed algorithm has a regret that scales inverse exponentially with the length of the lookahead for perturbations. In addition, the paper shows that a natural strategy (MPC) has a performance ratio that goes to one given a lookahead of log T, against the best online algorithm. While the latter is purely an computational question, its import is due to the fact that for adversarial w's, the optimal offline strategy is not clearly (it is unknown) efficiently computable.

Strengths: + The paper studies a problem that is practically relevant. An online high-level trajectory planning algorithm often outputs some states in to the future for the low-level controller to track. + The algorithmic solution (MPC) is nice even as the optimal online strategy for the stochastic case is efficiently computable. This is so because the suggested algorithm can use a black-box LQR solver.

Weaknesses: + The reliance on exact prediction (as noted in the paper) is unfortunate, and casts doubts on the robustness of the approach.

Correctness: The claims made, while not thoroughly checked, seem plausible.

Clarity: The paper is generally well written.

Relation to Prior Work: As far as the reviewer is aware, rigorous bounds for this setting are unavailable in prior art.

Reproducibility: Yes

Additional Feedback: Thanks for the discussion -- the score is retained.


Review 3

Summary and Contributions: -- The paper studies the problem of linear quadratic regulator (LQR) control in settings where the system knows (exactly) the future disturbances for a window of time. The paper discusses two types of disturbances, namely, stochastic and adversarial disturbance. The authors provide lower bounds on the dynamic regret in both cases. They also analyze the performance of the model predictive control (MPC) algorithm and based on that, derive upper bounds on the dynamic regret. ---- UPDATE ---- I want to thank the authors for their comprehensive response to my comments. In particular, adding the points made regarding the motivation for having access to exact disturbances and adding the results with inexact disturbances will make the paper stronger.

Strengths: -- Model predictive control is a widely used algorithm in practice which relies on having access to a reliable model of the system in a finite horizon ahead. The paper’s analysis of the dependence of the regret on the length of this horizon is indeed interesting to the control and learning communities. -- The authors focus on global optimal policies which may be adaptive (as opposed to static policies) and are more challenging to analyze. In addition to dynamic regret, they also use the notion of performance ratio that is more realistic as it considers the same amount of information for an algorithm and the optimal algorithm. -- In contrast to a bulk of existing work, the authors do not restrict the optimal controller to be a linear function. -- The theoretical analysis of the paper seems sound.

Weaknesses: -- I think that the paper needs more motivation to justify accessibility to exact disturbances. The disturbance in the LQR model is capturing the unknown part of the model, and hence, exact access to that even for a limited window needs a deeper discussion. In fact, the unavailability of such exact model in many realistic applications has advanced the robust MPC framework. -- The paper lacks empirical evaluations. While this is not atypical for papers focusing on theoretical regret analysis, having empirical evaluations can reinforce the results.

Correctness: -- The methodology and the theoretical claims of the paper seem correct.

Clarity: -- Overall, the writing is satisfactory. -- The statement of theorems and corollaries are somewhat informal. These statements need to be formal, accompanied with descriptive and technical sentences, and (almost) self-contained. -- I suggest the authors to add more explanation to the steps of the proofs in the supplementary material. -- Titled paragraphs should only capture one paragraph. If there is more than one paragraph, subsections would make the structure clearer.

Relation to Prior Work: -- The authors have mentioned the relevant literature and clearly distinguished their work.

Reproducibility: Yes

Additional Feedback: -- “MPC” in the abstract has not yet been defined. -- Line 46: Please rephrase “dynamic regret minimizing policies” -- Algorithm 1: No need to have a second “Input” within the loop -- Line 145-146: “\phi” is not defined explicitly in Algorithm 1 -- Line 168-172: I suggest the authors to give more intuition -- Line 203-204: Be more specific than “later in the paper” -- Line 211: Be more specific about which part of the analysis is novel -- Line 219: Why “\tilde{Q}_f = P”? Isn’t “\tilde{Q}_f” given? -- It would help to add a table summarizing the results. If more space is needed, some of the results may be taken to the supplementary material. -- Line 270: Could you give some intuition why the same order of regret is attainable for the adversarial case? -- When describing the notation, please describe the difference between the notations used for the order (“O” vs. “\Theta” vs. “\Omega”). -- In the bibliography, the year of some references have been repeated. -- Line 470: Avoid saying “turns out to be correct” -- Proof of Theorem 4.4 does not discuss the result for Example 4.3. -- Line 594: Where is “\Omega” used in the proof?


Review 4

Summary and Contributions: This paper investigates the effect of predictions in online Linear Quadratic Regulator control with stochastic and adversarial disturbances. Upper bounds for the optimal cost and minimum dynamic regret is computed given access to predictions. The model uses general process disturbances with only stabilizability assumption.

Strengths: The paper enjoys the merit of rigorous derivations and theoretical analysis of the proposed results. The problem statement and description of the model is clearly written and the theoretical results are rigorously established. Performance bounds are and relative benefit of additional predictions are calculated.

Weaknesses: I failed to understand the motivation and originality of the work. In particular, the strict improvement in performance for access to predictions of future disturbances is not quite clear to me, specially in the context of exponential decay of benefit given added predictions. The literature for stochastic model predictive control (SMPC) is quite vast and the claim of originality for general additive disturbance and adversarial setting needs to be supported by more concrete citations and possible comparisons with similar works, e.g., general additive disturbance: Daniel E. Quevedo, Debasish Chatterjee, Stochastic predictive control, International Journal of Robust and Nonlinear Control, 10.1002/rnc.4722, 29, 15, (4985-4986), (2019), Joel A. Paulson, Edward A. Buehler, Richard D. Braatz & Ali Mesbah (2020) Stochastic model predictive control with joint chance constraints, International Journal of Control, 93:1, 126-139, DOI: 10.1080/00207179.2017.1323351; Adversarial setting: Optimal Attack against Autoregressive Models by Manipulating the Environment, Yiding Chen and Xiaojin Zhu, arXiv: 1902.00202, 2019.

Correctness: I have not gone through every step of the proofs but overall the methods seem correct.

Clarity: The authors show effort in clearly explaining the model and the results. However, the merit and practicality of a system having access to predicted disturbances is not clear to me.

Relation to Prior Work: The authors made an effort to cite relevant background work. However, I believe the merit and originality of the paper can be enhanced by comparing works that are closer in spirit. I have mentioned a few in my previous comment and I am sure there are more such prior work that need to be cited.

Reproducibility: Yes

Additional Feedback:

[Author Response · NeurIPS 2020]

**General Response.** Since some comments overlapped, we first provide a general response before individual questions.

• Some reviewers mentioned the exact prediction assumption in the paper. Definitely the inexact prediction case is very
interesting and important, and we are currently working on that. However, we would like to emphasize:

a. Exact predictions are very common in online learning literature (e.g., [23], [24], [16]), where the focus is on how
future information improves algorithmic optimality (the metric is typically regret or competitive ratio).
b. Note that the disturbance $w_t$ in our model is much more general than just "noise" — in Section 4 we consider the
most general jointly stochastic $w_t$ (not necessarily i.i.d.) and in Section 5 we consider adversarial $w_t$. Our setting
supports a lot of real-world applications. For example, in the LQ tracking example in Section 2, predictions of $w_t$
are from desired trajectories, which are usually predefined and exact in robotics (as Reviewer 2 mentioned, *"An
online high-level trajectory planning algorithm often outputs some states in the future for the low-level controller
to track."*). In other words, the exact prediction assumption is actually practical in many scenarios, since our $w_t$ is
beyond unstructured "noise".
c. If the predictions are inexact, the exponentially decaying properties will be kept and we will have a residual term for
prediction errors (i.e., regret becomes $O((\lambda^k + \sum_{i=0}^{k-1} \epsilon_i \lambda^i)^2 T)$, where $\epsilon_i$ is the prediction error), according to our
current research results. We would like to add the inexact case since it is a straightforward add-on.

• Some reviewers asked about the relationship between our results and more general MPC settings, especially robust
MPC, learning unknown dynamics in MPC, and stochastic MPC. Some of these settings are indeed relevant to our work
or can be integrated with our work, so we will definitely cite more papers in this field and reorganize them (if accepted
we will have one more page to do that). However, we would like to emphasize:

a. In this paper, we adopt a conventional definition of MPC as an online deterministic optimal control problem with
a finite-time horizon with dynamics constraints. This type of MPC is the most natural and simple policy given
deterministic and finite future predictions — it is why we call it a "greedy and naive" policy in our paper. Definitely
the general MPC family is more sophisticated (e.g., considering constraints, robustness and safety), but our goal is to
show future predictions allowing simple algorithmic ideas to be effective and to connect online learning with control.
b. Completely different from robust MPC, learning unknown dynamics in MPC and stochastic MPC settings, this paper
focuses on **optimality** and uses standard metrics in the online learning community (dynamics regret and performance
ratio). We focus on the optimality gap between MPC, the optimal online policy and the optimal offline policy (with
the knowledge of all future) in this paper. On the other hand, other settings in MPC mentioned by R1 and R4 focus
on robustness, stability and safety. As mentioned by R2 (*"As far as the reviewer is aware, rigorous bounds for this
setting are unavailable in prior art."*), this paper is the first result that guarantees non-asymptotic optimality of the
most standard MPC policy. We will make this comparison clearer and cite these papers.
c. This paper not only analyses the performance of MPC, but also studies the **fundamental limit** given $k$ future exact
predictions (i.e., we characterized the optimal policies given $k$ predictions, and derived the regret lower bounds of
any online policy), which is not covered in the prior arts.

**Reviewer 1:** Thanks for your constructive feedback and pointing out related work about learning to MPC. We hope the
general response helps, and we would definitely reorganize related literature and add more systematical discussions.

**Reviewer 2:** Thanks for your valuable comments and suggestions. We hope the general response addresses your
concern about exact predictions.

**Reviewer 3:** Thanks for your constructive reviews. We hope the general response addresses your concern about the
exact predictions. Regarding empirical studies, we would like to add some motivating numerical examples in the main
body since we will have one more page if accepted. For technical questions:

*Line 168-172: I suggest the authors to give more intuition.* The key idea is that $\max(f - g) \geq \max f - \max g$.

*Line 219: Why $\tilde{Q}_f = P$?* The terminal cost $Q_f$ is given, but the virtual terminal cost $\tilde{Q}_f$ used in MPC is to be designed.

*Line 594: Where is "$\Omega$" used in the proof?* When reducing the adversarial case to the stochastic case, we require that
the support of the distribution of each $w_t$ is contained in $\Omega$.

*The difference between $O()$, $\Omega()$ and $\Theta()$.* We say $f(k) = O(g(k))$ if $\exists C > 0, \forall k \geq 1, |f(k)| \leq C\, g(k)$; $\Omega()$ is similar
except that the last "$\leq$" is replaced by "$\geq$"; $\Theta()$ means both $O()$ and $\Omega()$. This is stronger than the standard definition
where $f(k) = O(g(k))$ if $\exists C > 0, k^* > 0, \forall k \geq k^*, |f(k)| \leq C\, g(k)$. We will make it clearer in the paper.

**Reviewer 4:** Thanks for your constructive reviews and we hope the general response addresses your concern about the
relationship with other MPC settings and the practicality of predicted disturbances.

The papers you mentioned have studied MPC in various settings, but they demonstrate their effectiveness mainly
through empirical ways. Some papers have theoretical results for stability, but optimality guarantees are lacking. As
far as we know, our paper is the first one to provide theoretical guarantees for the dynamic regret of MPC. We will
definitely make this comparison clearer, and add more references and discussions.

[Meta-Review · NeurIPS 2020]

Reviewers 1-3 were enthusiastic about the technical results in this paper and are in favor of accepting. While reviewer 4 was less enthusiastic, they did not defend their position during discussion. All of the reviewers felt that the paper can be improved substantially by including more / better comparison with the broader literature on MPC. Based on the rebuttal, it sounds like the authors are taking steps towards doing this, and this should be included in the final version of the paper.